# Structural insights into the mechanism of DNA branch migration during homologous recombination in bacteria

Leonardo Talachia Rosa [1,3,5], Émeline Vernhes [2,4,5], Anne-Lise Soulet[2], Patrice Polard [2✉] & Rémi Fronzes [1✉]

## Abstract

**Some DNA helicases play central and specific roles in genome maintenance and plasticity through their branch migration activity in different pathways of homologous recombination. RadA is a highly conserved bacterial helicase involved in DNA repair throughout all bacterial species. In Gram-positive Firmicutes, it also has a role in natural transformation, while in Gram-negative bacteria, ComM is the canonical transformation-specific helicase. Both RadA and ComM helicases form hexameric rings and use ATP hydrolysis as an energy source to propel themselves along DNA. In this study, we present the cryoEM structures of RadA and ComM interacting with DNA and ATP analogs. These structures reveal important molecular interactions that couple ATP hydrolysis and DNA binding in RadA, as well as the role of the Lon protease-like domain, shared by RadA and ComM, in this process. Taken together, these results provide new molecular insights into the mechanisms of DNA branch migration in different pathways of homologous recombination.**

**Keywords** DNA Recombination; Helicase; DNA Translocation; Bacterial Transformation
**Subject Categories** DNA Replication, Recombination & Repair; Microbiology, Virology & Host Pathogen Interaction; Structural Biology

## Introduction

Homologous recombination (HR) is a conserved mechanism of DNA strand exchange that is central to genome biology in all living cellular organisms (bacteria, archaea and eukaryotes) and in some of their viruses. Many different HR pathways are key to common and essential cellular processes such as DNA repair and replication, meiosis in eukaryotes, or horizontal gene transfer in prokaryotes.

Any failure in these pathways can have dramatic consequences, threatening the viability or genetic integrity and evolvability of the cell, which can lead to many pathological disorders such as cancer or developmental defects (Kreuzer, 2005; Sun et al, 2020).

HR pathways proceed through successive steps, with some reactions common to all and others more specific. The initial HR steps consist of DNA strand exchange reactions that are invariably catalyzed by a conserved ATPase known as the HR recombinase, called RecA in bacteria, RadA in archaea, and Rad51 and Dmc1 in eukaryotes. This enzyme acts in the form of a polymer initially assembled on single-stranded DNA (ssDNA), known as the presynaptic HR filament. This first HR intermediate is capable of searching for a complementary sequence in a recipient double-stranded DNA (dsDNA) molecule. The recombinase then proceeds to pair the bound ssDNA molecule with its complementary sequence in the dsDNA, creating a branched DNA molecule (Hertzog et al, 2023; Yang et al, 2020). This second HR intermediate is called the synaptic or heteroduplex HR product and is most commonly referred to as the D-loop structure (Chen et al, 2008; Yang et al, 2020). Each HR pathway involves a distinct subset of effectors that act with or independently of the recombinase to orchestrate the formation of the presynaptic and synaptic HR intermediates and the D-loop maturation steps. A D-loop processing reaction common to many HR pathways is the subsequent pairing of the second DNA strand of the donor and recipient DNA molecules, on one or both sides of the D-loop, leading to the formation of a single or double Holliday junction (HJ or dHJ, respectively). The final outcome of each HR pathway is then determined by how these three- or four-stranded HR intermediates are processed. A key reaction is to extend or reverse the exchange of paired DNA strands by driving DNA branch migration, either at one or both boundaries of the D-loop or at each HJ. DNA synthesis may be involved in the maturation step of these different HR products. Finally, another maturation step of these HR intermediates consists in their specific cleavage to resolve and separate the joined DNA molecules (Sun et al, 2020; Yang et al, 2020).

A distinct D-loop maturation reaction that drives three-stranded DNA branch migration has recently been characterized in bacteria in the HR pathway of natural transformation (NT). This reaction is

[1]Structure and Function of Bacterial Nanomachines—Institut Européen de Chimie et Biologie, Microbiologie fondamentale et pathogénicité, UMR 5234, CNRS, University of Bordeaux, 2 rue Robert Escarpit, 33600 Pessac, France. [2]Laboratoire de Microbiologie et de Génétique Moléculaire (UMR 5100). Centre de Biologie Intégrative; 169, avenue Marianne Grunberg-Manago; CNRS—Université Paul Sabatier—31062, Toulouse Cedex 09, France. [3]Present address: Departamento de Bioquímica e Biologia Tecidual. Laboratório de Bioquímica de Complexos Bacterianos. Instituto de Biologia. Universidade Estadual de Campinas (UNICAMP), Monteiro Lobato, 255, Campinas 13083-862, Brasil. [4]Present address: TBI, Université de Toulouse, CNRS, INRAE, INSA, Toulouse, France. [5]These authors contributed equally: Leonardo Talachia Rosa, Émeline Vernhes.
✉E-mail: patrice.polard@univ-tlse3.fr; remi.fronzes@u-bordeaux.fr

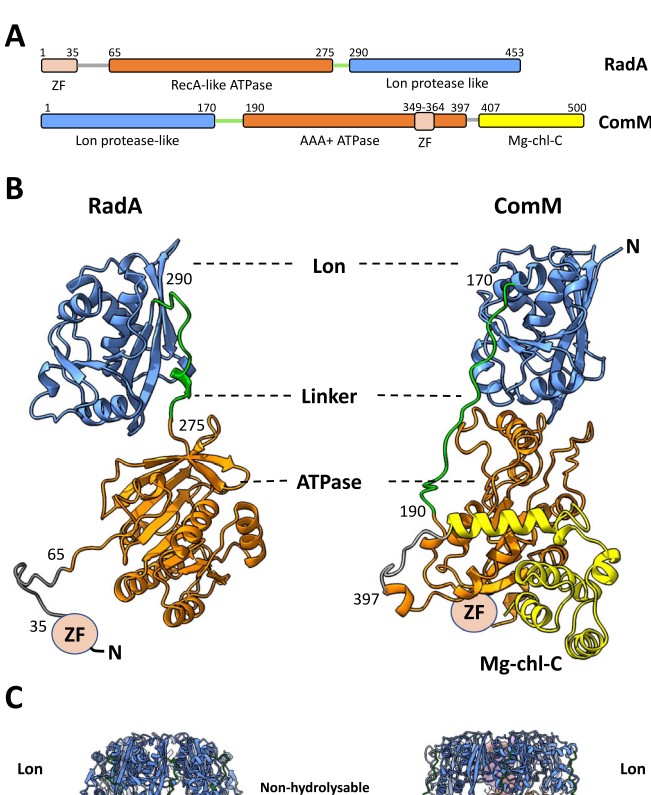

**Figure 1. Domain representation of RadA and ComM.**

(A) Sequence representation of RadA (top) and ComM (bottom), separated by domains, with the respective start and end residues of each domain labeled at the top. Lon protease-like domains are shown in blue, while ATPase domains (RecA-like and AAA+) are shown in orange. Linkers connecting the Lon and ATPase domains are shown in green. Predicted zinc finger domains are shown in pink. The C-terminal magnesium chelatase-like domain is shown in yellow. (B) Structure of RadA (left) and ComM (right), shown as ribbons and colored according to the sequence representation in (A). Both structures (subunit C) are aligned with their Lon domain in the same orientation. The zinc finger domain has not been resolved in either structure and is shown as a circular blob. The N-terminal extremities are indicated. (C) Ribbon representation of the RadA (left) and ComM hexamers, colored according to the sequence representation in (A). Non-hydrolysable nucleotides (ATP-γ-S for RadA and AMPPNP for ComM) are shown as a surface representation in magenta. dsDNA is shown as a surface representation in pink and light purple for each DNA strand.

positive Firmicutes (Nero et al, 2018). RadA is universally involved in HR-dependent DNA repair and is additionally involved in NT in the model transformable firmicutes species *Streptococcus pneumoniae* and *Bacillus subtilis* lacking ComM (Beam et al, 2002; Torres et al, 2019). In contrast, ComM has been shown to be exclusively involved in NT in other transformable species, including the Gram-negative species *Vibrio cholerae* (Nero et al, 2018). Comprehensive biochemical characterization of RadA from *Escherichia coli*, *S. pneumoniae*, *B. subtilis* and *Thermus thermophilus* and ComM from *V. cholerae* revealed that they are functional homologs during HR (Cooper and Lovett, 2016; Marie et al, 2017; Torres et al, 2019; Inoue et al, 2017; Nero et al, 2018). Both RadA and ComM were found to be ring-shaped hexameric DNA motors powered by ATP hydrolysis to direct three-stranded DNA branch migration in vitro. In addition, genetic analysis of their central role in NT of *S. pneumoniae*, *B. subtilis* and *V. cholerae* demonstrated that both promote elongation of ssDNA integration in this HR pathway.

RadA and ComM differ significantly at the structural level. RadA is composed of an N-terminal C4 zinc finger domain (ZF domain), a central RecA-like ATPase domain, and a C-terminal Lon protease-like domain (Lon domain) (Fig. 1A). RadA from *S. pneumoniae* forms hexamers in the absence of DNA and translocates on DNA in the 5' to 3' direction, as deduced from its helicase activity on various DNA substrates (Marie et al, 2017). In parallel with its biochemical analysis, we reported the crystal structure of *S. pneumoniae* RadA in complex with dTDP (Marie et al, 2017). In this structure, RadA forms a hexameric ring with C2 symmetry. While the Lon and ATPase domains were resolved, the ZF domain was not visible. The atomic structure of ComM is not known, but it is predicted from sequence analysis to contain an N-terminal Lon protease-like domain (Lon domain), a central AAA+ ATPase domain with an embedded zinc finger, and a C-terminal $Mg^{2+}$ chelatase domain (Mg-chl-C, Fig. 1A). ComM was described as a monomer in solution, forming hexamers in vitro only in the presence of ATP and DNA, and was shown to have bidirectional helicase activity (Nero et al, 2018).

While the role of RadA and ComM in DNA branch migration during HR is well established, little is known about the molecular basis of their interaction with DNA, including their functional assembly and ATP-dependent DNA translocation mechanism. Here, we present and compare the cryoEM (cryogenic electron microscopy) structures of RadA and ComM from *S. pneumoniae* and *Legionella pneumophila*, respectively, in interaction with DNA. Both proteins were found to be stably bound to dsDNA in the presence of slowly hydrolyzable ATP analogs, revealing the overall architecture of these two translocases dictated by their contacts with DNA strands. While RadA and ComM share the same hexameric scaffold formed by the Lon domains, they show substantial differences in the arrangement of the ATPase domains relative to the DNA and the Lon domain ring. These findings shed light on different mechanisms of ATPase activation by DNA and suggest a regulatory function of the Lon domain in these two hexameric motors that drive three-stranded DNA branch migration during HR.

## Results

### Cryo-electron microscopy reveals the conformation of RadA bound to DNA

RadA from *S. pneumoniae* was purified from *E. coli* mainly as a hexamer and shown to exhibit ATPase activity in the presence of DNA,

catalyzed by either RadA (not to be confused with archaeal recombinase) or ComM ATPases, depending on the species (Marie et al, 2017; Nero et al, 2018). NT consists of the uptake and internalization of exogenous DNA into the cell as linear ssDNA, which is then integrated into the genome by RecA-directed HR. This key biological process alters the genetic content of bacteria in a variety of ways. These include gene insertion and deletion, as well as multiple types of genomic rearrangements, all of which promote bacterial adaptation and evolution in response to stress, including antibiotic resistance and vaccine escape (Johnston et al, 2014). In contrast to RadA, which is widely conserved across bacterial species, ComM is less well-distributed, as notably absent in Gram-

with lower activity on dsDNA than on ssDNA (Marie et al, 2017). To gain a better understanding of the interaction between RadA and DNA, purified RadA was incubated in the presence of the poorly hydrolyzable ATP analog ATPγS with a gapped DNA substrate consisting of 30 base pairs (bps) of dsDNA followed by 60 nucleotides (nts) of ssDNA and 30 bp of dsDNA (Combination CD in Appendix Table S1). Density maps of hexameric RadA bound to DNA were obtained using cryoEM, yielding a resolution of 3.15 Å (Appendix Fig. S1A–F).

In the atomic model built within these maps, the overall protomer structure closely resembles the RadA crystal structure solved without DNA (PDB: 5LKM) (Marie et al, 2017) (Fig. 1B). In both the crystal and CryoEM structures, the central ATPase domains (residues 65–275) and the C-terminal Lon domains (residues 290–453) of each subunit of the hexamer are connected by a 15-residue linker (residues 275–290) and the N-terminal ZF domain could not be resolved (Fig. 1B,C).

In the CryoEM structure of DNA-bound RadA, the Lon domain has planar and hexameric ring-shaped C6 symmetry as in the DNA-free crystal structure of RadA. Strikingly, the ATPase domains adopt a helical configuration in the dsDNA-bound RadA structure (Figs. 1C and 2A). The flexible linker connecting the Lon and ATPase domains allows for this symmetry mismatch (Fig. 1B,C). Five of the six ATPase domains (Subunits A–E) are clearly defined in the density map and subunit F, which is located farthest from the Lon domain, exhibits lower resolution in certain areas (Appendix Fig. S1). As in all RecA-like ATPases, the ATP-binding site is located between adjacent domains. The map shows that five molecules of ATP-γ-S and one $Mg^{2+}$ ion are well positioned within the active site (Figs. 1C and 2A). In the cavity between subunits A and F, only a fuzzy density was visible, indicating partial nucleotide binding in this region. Each protomer comprises an N-terminal arm that encircles the adjacent ATPase domain on subunits A–E, as previously observed in the crystal structure of DNA-free RadA (Fig. 2A). The N-terminal ZF domain was not visible in the map, suggesting that it is highly flexible relative to the ATPase domain, even when bound to DNA.

A clear density of a 21 bp-long dsDNA is visible. The density begins within the RadA hexamer at the boundary between the Lon and ATPase domains and expands towards the ATPase domain and beyond the hexamer ring (Fig. 2A, left panel). As the DNA becomes less restricted and more flexible, the density of the DNA progressively decreases. RadA appears to bind to the dsDNA ends of the gapped DNA substrate with its ssDNA extending to the outside of the hexameric ring on the ATPase side. While the DNA backbone has a similar structure to that of B-form DNA in the ATPase region (Appendix Fig. S1), the base pairing toward the Lon domain is less clear due to the lower resolution of the map in this region.

## CryoEM structures of *Legionella pneumophila* ComM

ComM from *L. pneumophila* fused to a C-terminal His$_6$-tag was purified following its overexpression in *E. coli* through nickel affinity chromatography and elution as a single peak on a Superdex-200 10/300 gel filtration column (Appendix Fig. S3). We observed a 50 kDa monomer protein state (Appendix Fig. S3). This is in agreement with the results reported for the homologous ComM protein from *V. cholerae*, which was shown to hexamerize in vitro only in the presence of ATP and ssDNA (Nero et al, 2018). Here, we incubated purified *L. pneumophila* ComM protein with

AMPPNP and a synthetic oligonucleotide consisting of 60 bp of dsDNA and 60 nt of ssDNA (Appendix Table S1, Combination EG), which enabled us to obtain the cryoEM density map of ComM hexamers bound to DNA and AMPPNP at a resolution of 3.1 Å (Appendix Fig. S2). Local refinements were also carried out on half of the hexamer and the Lon domain, resulting in an improvement in the quality of the map in these regions (Appendix Fig. S4A–C).

The final atomic model of the ComM protomer (Fig. 1B) reveals that the N-terminal domain comprises a Lon protease-like domain, as deduced from its sequence. Its AAA+ ATPase domain is similar to those of archaeal and eukaryotic MCM/CMG helicases and bacterial proteins such as RavA and magnesium chelatase (Fig. EV1) (Baretić et al, 2020; Fodje et al, 2001; Jessop et al, 2020). In addition, the C-terminal $Mg^{2+}$ chelatase domain is closely linked to the ATPase domain (Fig. 1B). Densities for the zinc finger domain of ComM were visible in the cryoEM map (Fig. 2B), but the resolution was too low for accurate modeling. Notably, although ComM and RadA have an inverted order for the Lon and ATPase domains in their linear sequences, they share identical topology in their 3D structures (Fig. 1). The long linker (residues 170–190) between the domains in ComM allows such a large swap.

The ComM consists of a two-tier hexameric ring (Figs. 1C and 2B). One ring is formed by the Lon domains and the other by the ATPase domains, which are surrounded by $Mg^{2+}$ chelatase domains. These latter domains do not interact with each other in the ring (Fig. 1C). The ATPase ring remains nearly planar relative to the Lon domain, except for the DNA-binding loops (as presented below in "DNA binding to ComM" section). A density corresponding to a 24 bp-long dsDNA segment was distinctly visible throughout the hexamer. This B-form dsDNA segment follows the hexamer symmetry axis, in contrast to the tilted axis adopted by DNA-bound RadA (compare Fig. 2A and B).

Further processing revealed that the dataset could be divided into three populations. 53% of the particles correspond to the protein bound to DNA and AMPPNP, used to generate the structure described above, and presumably stabilized in a translocated state. In 23% of the dataset, ComM hexamers are observed in the absence of DNA (Appendix Fig. S5). The ATPase domains adopt a cracked-ring C2 symmetry in this structure, while the Lon domains remain unchanged in comparison to the DNA-bound conformation. This state may represent a transient complex before or after DNA loading.

In the remaining 24% of the particles, ComM hexamers interact with each other via their ATPase domains to form a dodecamer bound to the same molecule of dsDNA (Appendix Fig. S6).

## ATPase active site in RadA

In our previous study (Marie et al, 2017), we attempted to obtain a nucleotide-bound state of RadA by crystallizing it in the presence of dTTP. However, the nucleotide was hydrolyzed during the crystallization process, and as a result, the structure was obtained in the presence of dTDP. Here, we reveal a cryoEM structure of the RadA hexamer bound to dsDNA with five ATPase active sites occupied by ATP-γ-S and the last unoccupied (Figs. 2A and 3A). Using the C/D subunit interface as a model, R136 and M265 in subunit D and G255 and S256 in subunit C coordinate the positioning of the adenosine base in ATP-γ-S. This occurs at the end of the KNRFG motif (residues 251–255), a conserved motif

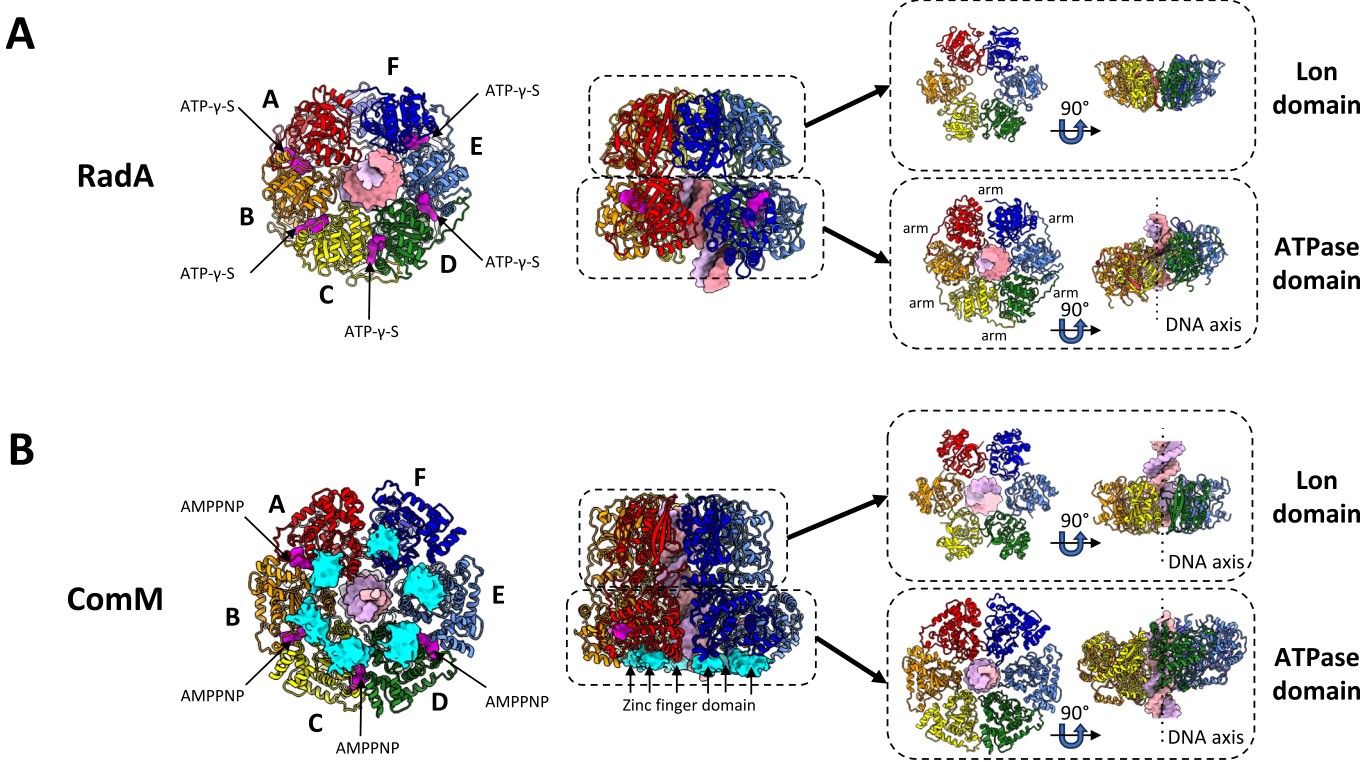

**Figure 2. Subunit representation and domain oligomerization in RadA and ComM.**

(A) From left to right. End view (from the ATPase domain) and side view of the RadA structure in ribbon representation. Subunits A, B, C, D, E, and F are colored red, orange, yellow, green, light blue, and navy blue, respectively. ATP-γ-S is shown as a surface in magenta. DNA is shown as a surface with each strand in pink and light purple. The oligomers formed by each domain (ATPase and Lon rings) are encircled and shown individually as end and side views in the inset. (B) From left to right. End view (from the ATPase domain) and side view of the ComM structure in ribbon representation. Subunits A, B, C, D, E, and F are colored red, orange, yellow, green, light blue, and navy blue, respectively. AMPPNP is shown as a surface in magenta. DNA is shown as a surface with each strand in pink and light purple. The oligomers formed by each domain (ATPase and Lon rings) are encircled and shown individually in the inset as end and side views. The electron density corresponding to the zinc finger domain is shown in cyan.

unique to RadA homologs (Marie et al, 2017). The Walker A motif (residues 95–103) in subunit D coordinates the α- and β-phosphates with R133. Sterically constrained on the opposite side is R253 of the KNRFG motif in subunit C, serving as an arginine finger to promote energy conservation during catalysis (Fig. 3A). Other interactions stabilize the γ-phosphate, including K251 from the KNRFG motif in subunit C. This specific residue is commonly termed as the "piston" for ATP cleavage and its mutation significantly hinders transformation in vivo (Marie et al, 2017). The catalytic glutamate, E124, in subunit D could coordinate a nucleophilic water molecule (not visible in our map) for ATP hydrolysis (Lyubimov et al, 2011). Further coordination is achieved by K101, and S102 from the Walker A motif of subunit D. This motif additionally coordinates with a $Mg^{2+}$ ion, as well as with E125 and D170 of subunit D (Fig. 3A). This description is in agreement with our previous results (Marie et al, 2017), which showed that mutants in the Walker A (K101A) and KNRFG (K251A and R253A) motifs were severely affected for ATPase activity.

## ATPase active site in ComM

The ATPase domains of the bacterial chaperone protein RavA (PDB 6SZB) and the archaeal MCM replication helicase (PDB

6SKL) are very similar to the ComM ATPase domain (Fig. EV1). The structure of ComM that is bound to dsDNA was obtained in the presence of the non-hydrolyzable ATP analog AMPPNP, which is located at the interface between two adjacent subunits (Fig. 3B). Densities for three AMPPNP molecules were observed at the interfaces between chains A and E, involving five protomers (Fig. 2B).

Taking the interface between chains C and D as a model, the adenosine nucleotide is coordinated by M226 and K195 of subunit C (Fig. 3B). The β-phosphate is surrounded by residues 221–225 of subunit C Walker A (residues 218–225). The γ-phosphate is coordinated by the side chains of K224 (lysine piston), T225, and E302 of subunit C. The catalytic glutamate, E302, in subunit C could coordinate a nucleophilic water molecule (not visible in our map) for ATP hydrolysis. In addition, the side chains of R316 (sensor 3 motif) and R463 (arginine finger) of subunit D are also involved in the coordination.

## DNA binding to RadA

In the crystal structure of RadA without DNA, the ATPase domains organize in a planar hexameric ring (Marie et al, 2017). Upon interaction with dsDNA, the ATPase domains adopt a

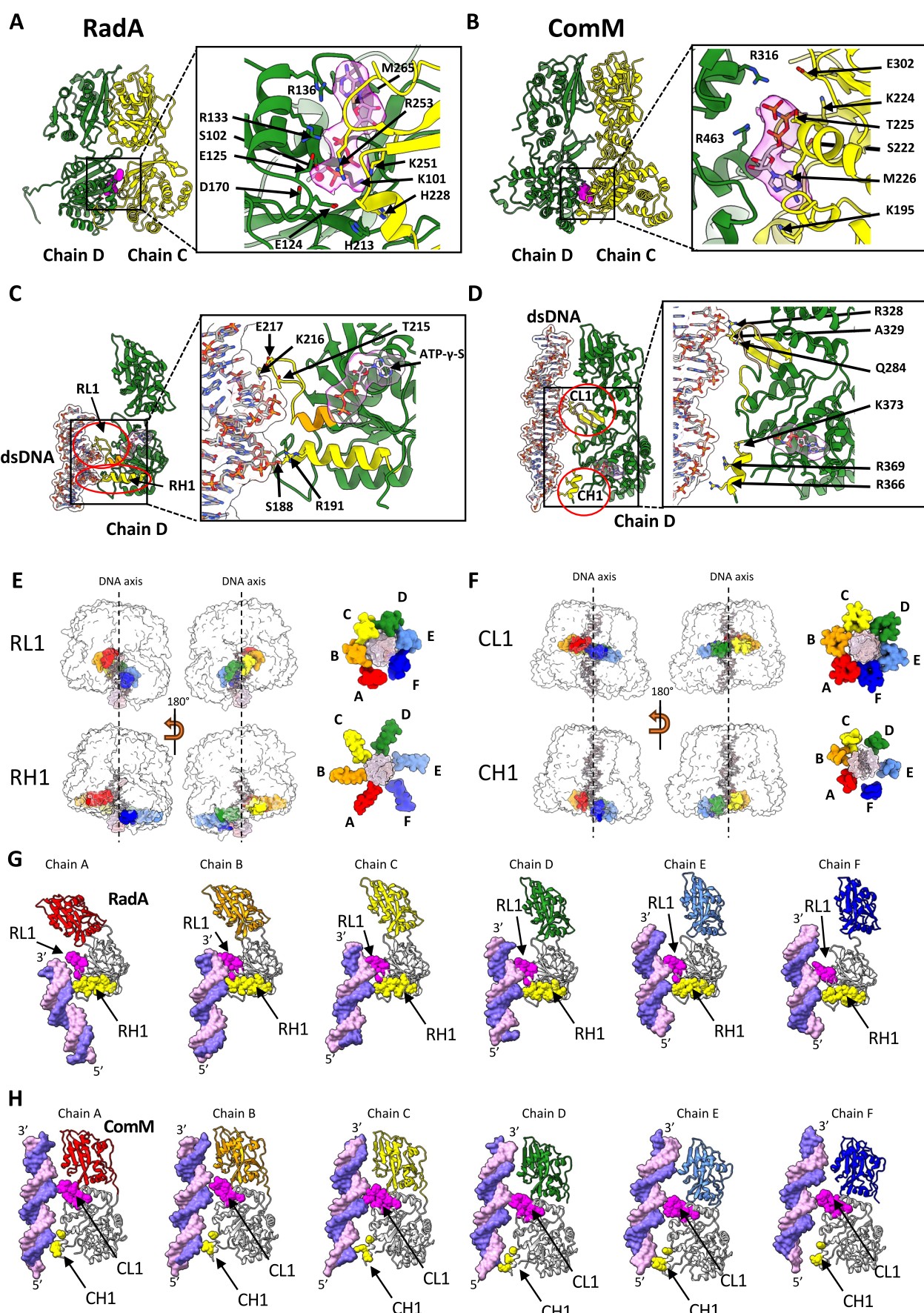

◄ **Figure 3.  ATPase active site and DNA-binding loops in RadA and ComM.**

(**A**) Active site of the ATPase domain in RadA. The left panel shows ribbon representation of subunits D (green) and C (yellow). ATP-γ-S is shown in magenta surface representation. The right panel shows a zoom of the ATPase active site with subunits D and C in ribbon representation. ATP-γ-S is shown in stick representation. The electron density corresponding to ATP-γ-S in the CryoEM map is seen as a transparent area. Residues involved in ATP-γ-S binding and/or catalysis are shown as sticks. $Mg^{2+}$ ion is shown as a red sphere. (**B**) Active site of the ATPase domain in ComM. Left panel shows ribbon representation of subunits D (green) and C (yellow). ATP-γ-S is shown in magenta in surface representation. The right panel shows a zoom of the ATPase active site with subunits D and C in ribbon representation. ATP-γ-S is shown in stick representation. The electron density corresponding to ATP-γ-S in the CryoEM map is seen as a transparent area. Residues involved in ATP-γ-S binding and/or catalysis are shown as sticks. (**C**) DNA-binding sites in RadA. Left panel shows ribbon representation of subunit D (green). The DNA is shown in a stick representation. The RL1 and RH1 regions are encircled in red and depicted in yellow. The short helix at the N-terminus of the KNRFG motif (R224 to H228) is depicted in orange. Right panel shows a zoom of the DNA-binding site with key residues in stick representation. (**D**) DNA-binding sites in ComM. Left panel shows ribbon representation of subunit D (green). The DNA is shown in a stick representation. The CL1 and CH1 regions are encircled in red and depicted in yellow. Right panel shows a zoom of the DNA-binding site with key residues in stick representation. (**E, F**) The RL1 and RH1 loops of RadA (**E**) or the CL1 and CH1 loops of ComM (**F**) colored by chain (same color code used in Fig. 2) are shown as surfaces. DsDNA is shown as surface. The left part of the panel shows two side views of these loops, with the rest of the RadA structure shown as a transparent surface. The right part of the panel shows end views (as seen from the Lon ring) of the loops and DNA. (**G**) Interaction of RadA subunits with DNA. RadA subunits are shown as ribbons. The Lon domains are colored according to the color code used in Fig. 2. The ATPase domains are colored light gray, except for RL1 in magenta and RH1 in yellow. The dsDNA is shown as a surface. The 5' and 3' ends of the pink DNA strand are marked. The other DNA strand is colored purple. (**H**) Interaction of ComM subunits with DNA. ComM subunits are shown as ribbons. The Lon domain is colored according to the color code used in Fig. 2. The ATPase domains are colored light gray, except for CL1 in magenta and CH1 in yellow. The dsDNA is shown as a surface. The 5' and 3' ends of the pink DNA strand are marked. The other DNA strand is colored purple.

helical organization (Fig. 2A). This places the KNRFG motif, which includes both the arginine finger and the catalytic piston, in close proximity to the bound ATP molecule of the neighboring subunit (Fig. 3A). At the bottom of the ATP-binding pocket, the γ-phosphate is bound to H228 in subunit C, while H213 in subunit D could align E124 to activate a water for hydrolysis (Fig. 3A). This location is consistent with the positions of H465 and Q494 in the T7 Gp4 helicase, a close structural homolog of RadA, which coordinates the binding of a nucleophilic water molecule with ATP hydrolysis to facilitate DNA binding (Fig. EV2) (Gao et al, 2019). In RadA, we propose that the stacking of these two histidines could structure a short helix at the N-terminus of the KNRFG motif (R224 to H228), which in turn would push the adjacent loop (RadA loop 1 or RL1, residues 213–223) forward to promote DNA binding (Fig. 3C). RL1 includes DNA-binding regions found in the D1 and D2 loops of the T7-gp4 structure but lacks the extended loop region found in its homolog (Fig. EV2D), a distinctive feature that would be likely compensated for by additional DNA contacts with the Lon domain region (Inoue et al, 2017). Indeed, three residues (R305, R314, and K345) in the Lon domain of RadA from *T. thermophilus* (corresponding to K320, N329, and K360 in RadA from *S. pneumoniae*) are located on the inner surface of the Lon ring and could directly mediate DNA binding (Inoue et al, 2017).

The primary contact with DNA is established by K216, which binds to the phosphate of the DNA backbone and coordinates the 5' to 3' DNA strand (ZF to Lon domain polarity) in a manner similar to that of DnaB loops (Gao et al, 2019). In addition, T215, E217, and G222 are involved in coordination (Fig. 3C). A second region corresponding to the N-terminus of the helix 188–204, named RH1 (RadA helix 1), coordinates the complementary DNA strand. S188 and R191 attach to the DNA backbone six base pairs below the base pair coordinated by RL1 towards the ZF domain, when the DNA helix completes a half turn on its axis and the other strand is exposed to the corresponding RadA subunit (Fig. 3C; Movie EV1).

The helical ring topology of the ATPase domains mirrors the topology of DNA (Fig. 3E). RL1 and RH1 interact with DNA two nucleotides above their adjacent domain. Subunit B binds to DNA upstream (towards the Lon domain) while subunit E binds to DNA downstream (Fig. 3E; Movie EV1). RL1 in subunit A is slightly displaced and dissociated from DNA, showing that ATP hydrolysis leads to DNA release (Fig. 3E,G). All other subunits appear to interact with DNA in the same way as subunit D, with the exception of RH1 in chain F, which could be dissociated from DNA (Fig. 3E).

## DNA binding to ComM

The resolution of the ComM map is not sufficient to accurately identify the protein's contacts with dsDNA. In MCM members of the helicases superfamily 6, DNA binding occurs through two conserved hairpins directed toward the center of the pore. These are the pre-sensor I β hairpin (Ps1β) and the helix 2 insert (H2i) (Fig. EV1). ComM maintains the Ps1β loop from residues 324 to 336, with R328 that could form a crucial hydrogen bond with the DNA phosphate backbone and the side chains of A329 and A330 creating a hydrophobic surface (Figs. 3D and EV1). In ComM, the H2i loop, which consists of residues 281 to 289, is half the size of its MCM counterpart and prevents the extended hydrophobic interaction along the DNA, instead making a potential contact via a hydrogen bond with Q284 (Fig. 3D). Both loops, called CL1/2 for ComM loop 1 and loop2, do not follow the helical symmetry of the DNA like for RadA but still are arranged in a subtle right-handed helical pattern (Fig. 3F; Movie EV2). Although the ATPase domains of ComM also assemble in a subtle helical pattern, they remain almost planar in relation to the Lon domain. The flexibility of the CL1/2 hairpins enables them to arrange themselves on DNA in a helical manner, creating an uninterrupted hydrophobic chamber while leaving the rest of the domain undisturbed (Fletcher et al, 2003) (Fig. 3F). A second region of the protein makes further contact potentially via R366, R369, and K373 residues (Fig. 3D). As stated above, the lack of sufficient resolution in this region of the map prevents an accurate positioning of the side chains of the CL1 and CH1 loops. However, it appears that the CL1 loop points towards the 5'->3' DNA strand (ATPase domain -> Lon domain direction) in chains B, C, D, E and F. This loop could be dissociated from DNA in chain A. On the other hand, CH1 appears to interact with the DNA in chains A, B, C, D, E but is clearly dissociated from DNA in subunit F (Fig. 3F,H).

## RadA and ComM share the same Lon protease-like scaffold

RadA and ComM both contain a Lon protease-like domain, although they belong to different ATPase families. In RadA, the domain is situated in the C-terminal region, whereas in ComM it is located in the N-terminal region (Figs. 1A,B and 4A). Due to the extended linker connecting the Lon and ATPase domains in ComM, the topology of the Lon in relation to the ATPase domain is conserved in both proteins (Fig. 1B,C). Despite their low sequence identity (28%), the two Lon domains superimpose with a root mean square deviation (RMSD) of only 1 Å (Fig. 4B). In both DNA-bound states of RadA and ComM, as well as in the RadA crystal structure (PDB 5LKM) and in DNA-free ComM (Appendix Fig. S5), the Lon domain forms a planar hexamer with similar pore diameters (17.7 Å in RadA and 23.1 Å in ComM) (Fig. 4A).

The PDB-PISA server measures a significant difference in the buried surface area between adjacent subunits of the two proteins, with 678 Å$^2$ for ComM and 1028 Å$^2$ for RadA (Appendix Table S2). The interface between the subunits shows polar contacts, which may help to stabilize the hexamer arrangement. RadA shows electrostatic interactions between E306 and R400 and K354 and E367 (Fig. 4D). In ComM, there are electrostatic interactions between S26 and R50 as well as E22 and R117 (Fig. 4D).

In contrast to traditional Lon proteases, the interior of the Lon domain ring is rich in basic residues, consistent with potential DNA coordination at the center of the pore (Fig. 4C). This is further supported by the observation that ComM interacts with DNA via three lysines, K43, K46, and K64, which form hydrogen bonds with the DNA backbone within the central pore (Fig. 4E). Consistent with this, it has been reported that the Lon domain of RadA from *T. thermophilus* displays DNA-binding activity and that mutations in three residues directly facing the center of the pore, K320, N329, and K360 in RadA from *S. pneumoniae*, alter the DNA-binding ability of full-length RadA (Fig. 4E) (Inoue et al, 2017).

## In RadA, molecular interactions between the Lon and ATPase domains are critical for coupling ATP hydrolysis to DNA binding

In *T. thermophilus* RadA, two surface residues, R286 and R385 (R301 and R400 in *S. pneumoniae* RadA), play a critical role in the DNA-mediated ATPase and DNA-binding activities (Inoue et al, 2017). R400 is located at the interface between the RadA Lon domains, as discussed above. The role of R301 was not clear. This residue is located at the interface between the Lon and ATPase domains. We hypothesized that this residue might regulate the ATPase domain of RadA, rather than directly affecting DNA binding.

While the ATPase domains adopt a helical conformation in the DNA-bound RadA hexamer, the Lon domains remain planar. Thus, R301 interacts with different regions of the ATPase domains depending on the topology of each subunit (Fig. 5A). In subunits A and B, where the ATPase domain is located closer to the Lon domain, R301 appears to form a salt bridge with E76. Meanwhile, in subunits D and E, where the ATPase domains are farther from the Lon domain, the density map clearly shows an interaction between R301 and E239 (Fig. 5A). However, in subunit F, which is unbound to DNA, the ATPase domain is too distant to contact

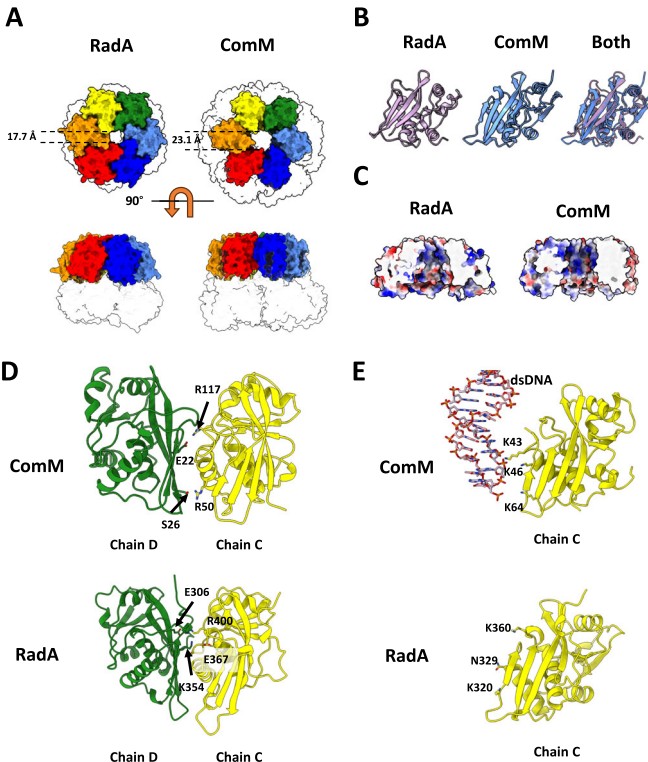

**Figure 4.  The Lon domain of RadA and ComM.**

(A) The Lon domains of RadA (left) and ComM (right) are shown as surfaces and colored according to the color code used in Fig. 2. Contoured white areas represent the rest of the protein. Top row, end view from the Lon ring side of the protein. Bottom row, side view. DNA is not shown. (B) Ribbon representation of the Lon domain of RadA and ComM (in purple and blue, respectively). A superposition of both is shown on the right panel. (C) Surface view of the Lon ring of RadA and ComM, colored according to the charge. The front part of the ring is cut out to show the inner surface. (D) Ribbon representation of the Lon domain of subunits C (yellow) and D (green) of ComM and RadA. Important residues at the interface between subunits are shown as sticks and marked. (E) Ribbon representation of the Lon domain of subunit C of ComM and RadA. Important residues at the interface with DNA (shown as a sticks) are shown as sticks and marked.

R301. Thus, we propose that R301 helps stabilize the different conformational states of the ATPase domain.

To evaluate the functional role of these molecular interactions, we attempted to purify site-directed alanine mutants of residues R301 and E239, which form this bridge in RadA from *S. pneumoniae*. The R301A site-directed mutant of RadA was successfully purified as wild-type RadA in the form of a soluble hexamer, but the E239A mutant was found to precipitate immediately upon purification, precluding its biochemical analysis. Therefore, we studied only the R301A mutant, starting with the analysis of its DNA-binding properties by electrophoretic mobility shift assay (EMSA), as previously done with wild-type RadA (Fig. 5B) (Marie et al, 2017).

RadA$_{R301A}$ has a significantly lower affinity for both ss- and dsDNA (Fig. 5B). This suggests that the RadA binding to DNA is dependent on the stabilization of the ATPase domain through the R301-E76/E239 bridges. The ATPase activity of the R301A mutant was then analyzed in the absence and presence of ssDNA. The

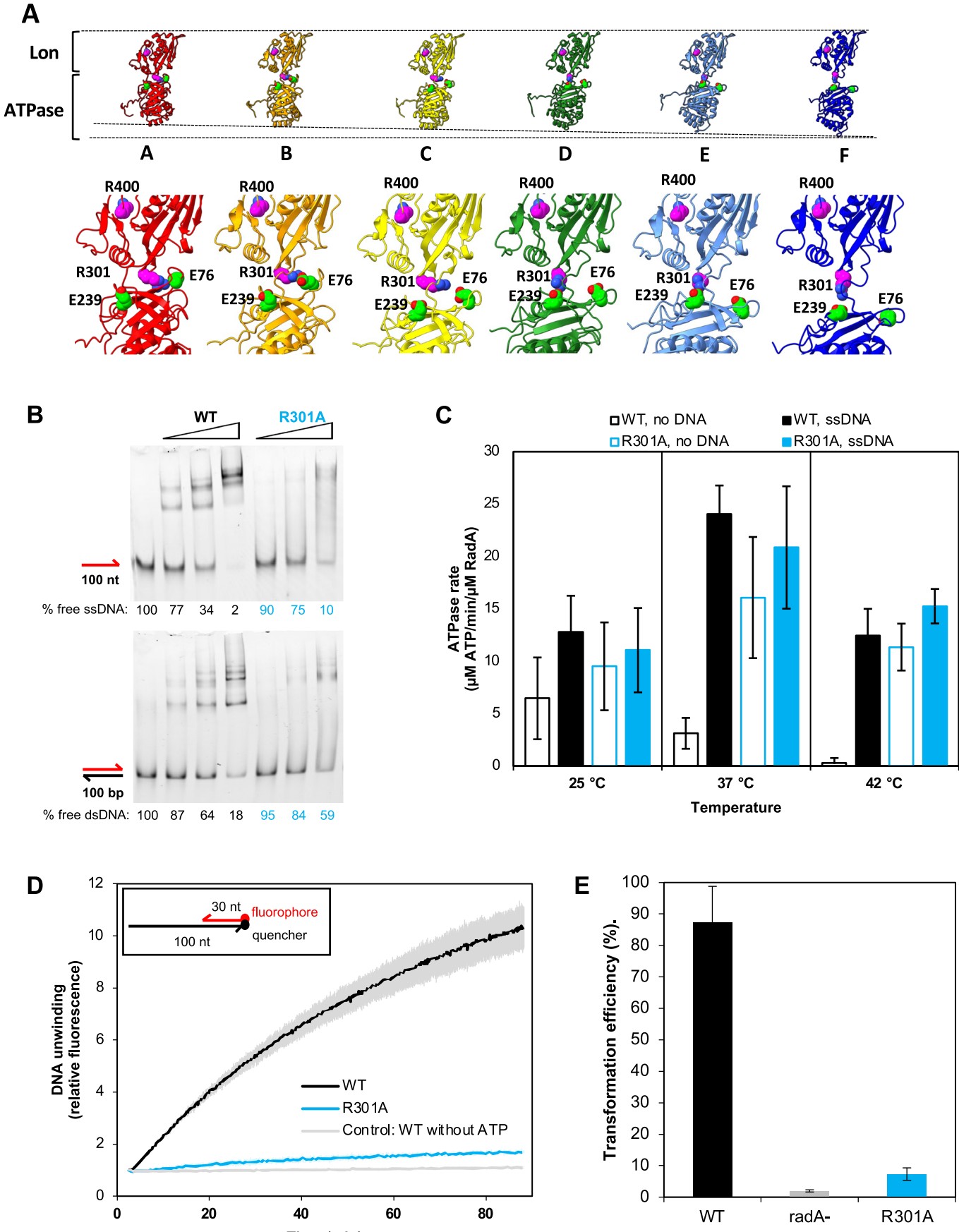

**Figure 5.   Salt bridges during RadA activity.**

(A) RadA subunits are represented side-by-side in equivalent orientation. Protomers color follow the same scheme as in Fig. 2. R301 and R400 densities are depicted in magenta, with nitrogen atoms in blue. E76 and E239 densities are depicted in green, with oxygen atoms in red. Along the translocation cycle, the ATPase domain moves farther from the Lon domain, accompanied by a subtle rotation movement. This rotation results in a salt bridge formed by R301, in the Lon domain, and E76 (subunits A, B, and C) and E239 (subunits D and E), in the ATPase domain. (B) DNA-binding properties of wild type and mutant RadA by Electrophoretic Mobility Shift Assay (EMSA). Binding to single-stranded DNA is shown in the upper panel, while binding to double-stranded DNA is shown in the lower panel. Free DNA migration position is indicated by the respective DNA depiction on the left (fluorescent DNA in red, unlabeled DNA in black), while protein-bound DNA has a delayed migration. DNA is incubated with incrementing quantities of protein, according to the gradient bar shown on top. Quantification of the free DNA bands is shown under each gel as a percentage of the first lane (no RadA) band. (C) ATPase activities at 25, 37, and 42 °C, in the absence or presence of single-stranded DNA. Error bars represent standard deviation and are calculated from at least three experimental replicates. (D) Helicase activities. DNA unwinding is shown as fluorescence increases over the initial fluorescence. A scheme of the DNA substrate used for the helicase assays is presented in the inset. The curves represent the mean of two experimental replicates. Error bars represent standard deviation and are shown as faded colors. (E) In vivo transformation efficiencies in *S. pneumoniae*. Error bars are calculated from at least five experimental replicates. Source data are available online for this figure.

R301A mutant exhibited high levels of basal ATPase activity in the absence of DNA at 37 °C, suggesting that ATP hydrolysis is decoupled from its interaction with DNA and ATP is consumed in a futile cycle (Fig. 5C). This effect was partially mitigated when the experiments were conducted at a lower temperature (25 °C), but it was still evident at higher temperatures (42 °C). These results highlight a thermal effect on the coupling between ATP hydrolysis and DNA binding. To further investigate the decoupling between ATP cleavage and DNA binding in the RadAR301A mutant, we analyzed its helicase activity. The assay involved hybridizing a 30 nt-long DNA strand labeled with a fluorophore at its 5' end to the 3' region of a 100 nt strand labeled with a quencher at its 3' end. The spatial proximity of the two labels causes fluorescence quenching, resulting in low fluorescence intensity. When the two strands dissociate due to RadA helicase activity, the labels move apart, and fluorescence increases (Fig. 5D). Consistent with the ATPase and DNA-binding assays, the RadA$_{R301A}$ mutant exhibits very low helicase activity (Fig. 5D). Finally, we investigated the impact of disrupting the salt bridge between the Lon and ATPase domains of RadA on bacterial transformation activity (Fig. 5E). The RadA$_{R301A}$ mutant strain exhibited a 92% reduction in efficiency, comparable to the *radA-* null mutant (Burghout et al, 2007; Marie et al, 2017).

Taken together, these results highlight an important role for the salt bridges mediated by R301 from the Lon domain and E76/E239 from the ATPase domain in the catalytic ATP hydrolysis cycle of RadA modulated by DNA and inferred to reflect its translocation on DNA powered by ATP hydrolysis. Disruption of the R301-E76/E239 salt bridges can cause futile ATP consumption and alter DNA binding. This may be due to increased flexibility in the ATPase domain, preventing RadA from stabilizing the retracted or stretched conformations between the two domains. As a result, the RadA$_{R301A}$ mutant can still coordinate ATP binding and hydrolysis, but cleavage probably occurs before DNA binding is promoted and the helical assembly is established. Breaking the R301-E76/E239 salt bridges deactivates RadA DNA translocation and helicase activities, as well as its DNA branch migration activity on HR intermediates during NT.

## Discussion

In this study, we report the cryoEM structures of the DNA-bound hexameric DNA branch migration motors RadA and ComM. This

structural analysis revealed that these two bacterial HR effectors have a modular and similar topology. Both are organized as two-tiered hexamers, with one tier consisting of the ATPase domain ring and the second tier consisting of a Lon-like domain ring; this later domain distinguishes these two DNA motors from others known to be involved in DNA dynamics processes.

### The Lon domain acts as a common hexamerization module in RadA and ComM

Both RadA and ComM contain a Lon-like structural domain (Fig. 4). This domain has not been found in any other DNA-binding ATPases characterized to date, suggesting that it may confer specific properties to these two helicases in the context of HR.

We have previously proposed that the Lon domain of RadA acts as a scaffold for its hexamerization, which occurs in the absence of DNA and ATP (Marie et al, 2017). On the other hand, ComM hexamerization was mainly observed in the presence of ATP and DNA, and to a lesser extent in the presence of ATP alone, as reported for ComM from *V. cholerae* (Nero et al, 2018) and confirmed here for ComM from *L. pneumophila*. The PDB-PISA server displays a significantly different interaction surface between the Lon domains in the two proteins. RadA has a greater number of residues involved in interfacial interaction, resulting in a larger interfacial area and greater solvation-free energy upon hexamerization compared to ComM (−6.8 kCal/mol for RadA vs. −3.9 kCal/mol for ComM) (Appendix Table S2). In addition, the p-value linked to the free energy gain for RadA of 0.28 indicates a site-specific interaction between subunits. Conversely, the p-value for ComM, which is close to 0.5, reflects a non-specific interaction and is similar to the average value for its structure. These findings suggest that the hexamerization of ComM, which is dependent of DNA and ATP, is primarily driven by global solvation-free energy rather than by interactions between specific residues within the Lon domain.

### Possible mechanisms for RadA and ComM loading onto DNA

ComM is an AAA + DNA motor related to the archaeal MCM and eukaryotic MCM2-7 replicative helicases, whereas RadA is related to the bacterial DnaB replicative helicase (Nero et al, 2018; Marie et al, 2017). Their loading onto DNA at replication origins proceeds by different mechanisms, all of which depend on accessory

effectors. Thus, in vitro and in vivo loading of the eukaryotic replicative MCM2-7 helicase (SF6) onto DNA replication origins is mediated by the sequential intervention of the ORC system (origin recognition complex), Cdc6 and Cdt1. The helicase is assembled on DNA as a docedamer, linking two MCM2-7 hexamers through their N-terminal regions. Recruitment and loading of the *E. coli* DnaB helicase on DNA at the single chromosomal origin of replication (OriC) requires the action of DnaA and DnaC. DnaC binds to and breaks the homohexameric DnaB ring to mediate DnaB loading around one DNA strand (Arias-Palomo et al, 2019). RadA and ComM are both capable of self-loading on ssDNA and dsDNA in vitro. Whether they are capable of self-loading on their HR DNA substrates in vivo has not been established. We have previously reported evidence for a functional interplay between pneumococcal RadA and RecA at three-way DNA junctions, where RecA assists RadA helicase activity, raising the hypothesis that RecA could recruit RadA and/or mediate its loading on DNA (Marie et al, 2017). Potential interactions between ComM and other proteins in vivo are less documented. In the absence of DNA, purified RadA is predominantly in the form of a pre-assembled hexamer, whereas ComM is predominantly found in the form of a monomer. Therefore, while it is straightforward to envisage the oligomerisation of ComM around its DNA substrate from free subunits, it is more difficult to understand how DNA loading occurs for RadA, because of its hexameric configuration stabilized by the Lon domain observed in vitro. One possibility is that part of RadA is monomeric in vivo, which would allow its oligomerisation around DNA, as proposed for ComM. Another possible loading mechanism would be an opening of the RadA hexamer at the interface between two subunits to allow DNA encirclement. This could either be assisted by a protein partner and/or upon interaction with DNA, or be inherent to the dynamics of the DNA-free hexamer. In this regard, it is worth mentioning the recent cryoEM studies (Shin et al, 2020), which showed that Lon proteases form a stable hexamer in the presence of protein substrate, but in the absence of substrate form a "lock washer-like" cracked ring to facilitate substrate recognition and loading. Based on the structural similarity between the Lon protease domain and the Lon-like domain of RadA, this Lon-like domain may promote ring cracking in the presence of DNA to facilitate loading of the hexameric ring of RadA.

Finally, it is important to note that because we used short DNA fragments in this study, RadA could instead have been threaded along the DNA without the need to open the RadA ring during the process. However, we have previously shown that it is possible to obtain RadA hexamers loaded onto long circular ssDNA molecules in vitro (Marie et al, 2017), a process that requires the opening of the ring.

## Dodecamerization of ComM

In 24% of the cryoEM data, two ComM hexamers interact back-to-back through their ATPase domains, coordinating a single dsDNA molecule that encompasses both hexamers (Appendix Fig. S6). Dodecamers of the Superfamily 6 of helicases are well-known in the literature (Cheng et al, 2022; Miller et al, 2019; Noguchi et al, 2017). Eukaryotic and archaeal MCM helicases form inactive dodecamers when loaded on their DNA substrate at the origin of replication. Once activated, the two MCM hexamers split and run past each other to perform strand separation (Miller et al, 2019; Noguchi

et al, 2017). However, it is unclear whether the ComM dodecamer could act similarly during HR. The DNA encompassed by MCM hexamers usually presents a short bending at the interface between the two hexamers, which serves as a base for forking initiation. This bend is not observed in ComM hexamers, where the dsDNA maintains a classical B-form DNA conformation along its entire length. The MCM proteins interact through their N-terminal domains (NTD) and move past each other, with the NTD in the front, in the 3' to 5' direction. In addition, the NTD of MCM is topologically equivalent to the Lon protease domain of ComM. However, it appears that ComM dodecamerizes through the opposite interface, leaving the Lon domain at both extremities (Appendix Fig. S6). This suggests that ComM hexamers would not move past each other on ssDNA upon activation of their translocation as MCM hexamers, but rather away from each other on dsDNA (Fig. 6B).

## RadA and ComM DNA translocation mechanism

Both RadA and ComM form hexameric structures that encircle DNA in the presence of poorly hydrolyzable ATP analogs. We hypothesize that this conformation corresponds to their active translocation state on DNA. In RadA, the entire ATPase domain is arranged in a helical pattern around the DNA (Fig. 2A). The DNA structure facilitates the correct positioning of the ATP-binding pocket between two adjacent subunits (Fig. 3A). Upon ATP binding, the KNRFG motif, which includes the arginine finger and lysine piston that coordinate ATP hydrolysis, is repositioned. This in turn pushes the RL1 loop towards the DNA for interaction (Fig. 3C). We propose that ATP hydrolysis causes destructuration of the KNRFG motif, affecting adjacent residues and pulling the RL1 loop away from DNA binding. The ATPase domain is then free from the constraints imposed by ATP and DNA, allowing it to retract towards the Lon domain. Here, the KNRFG motif coordinates a new ATP molecule and interacts with DNA at a position 12 bp upstream of the previous site. This culminates in a 12 bp/6 ATP translocation cycle (Fig. 6A). In a previous study, we showed that disruption of the Walker A motif by introducing a K101A mutation into RadA did not significantly affect DNA binding. However, introducing a K251A or R253A mutation in the KNRFG motif resulted in reduced DNA binding (Marie et al, 2017). Based on the current model, this could be due to the inability of the KNRFG mutants to stabilize the necessary helical conformation required for DNA binding. Furthermore, based on previous site-directed mutagenesis work (Inoue et al, 2017), which found that mutations in different Lon domain residues affected DNA binding, we investigated the effect of R301 on RadA activity (R286 in *T. thermophilus* RadA). In the CryoEM structure, R301 is involved in a salt bridge with either E76 or E239 in the most retracted or stretched subunits, respectively (Fig. 5). Our biochemical assays indicate that disruption of this salt bridge results in a futile DNA-independent ATP hydrolysis cycle and almost completely abolishes helicase activity (Fig. 5C,D). In vivo, the R301A mutant shows a deficiency in natural transformation equivalent to that of a radA null mutant (Fig. 5E). The salt bridge between the Lon and ATPase domains may regulate RadA translocation activity by stabilizing either the retracted or extended subunit forms and blocking ATPase activity in the absence of DNA.

In ComM, the helical arrangement is restricted to the DNA-binding regions, while the surrounding regions of the ATPase

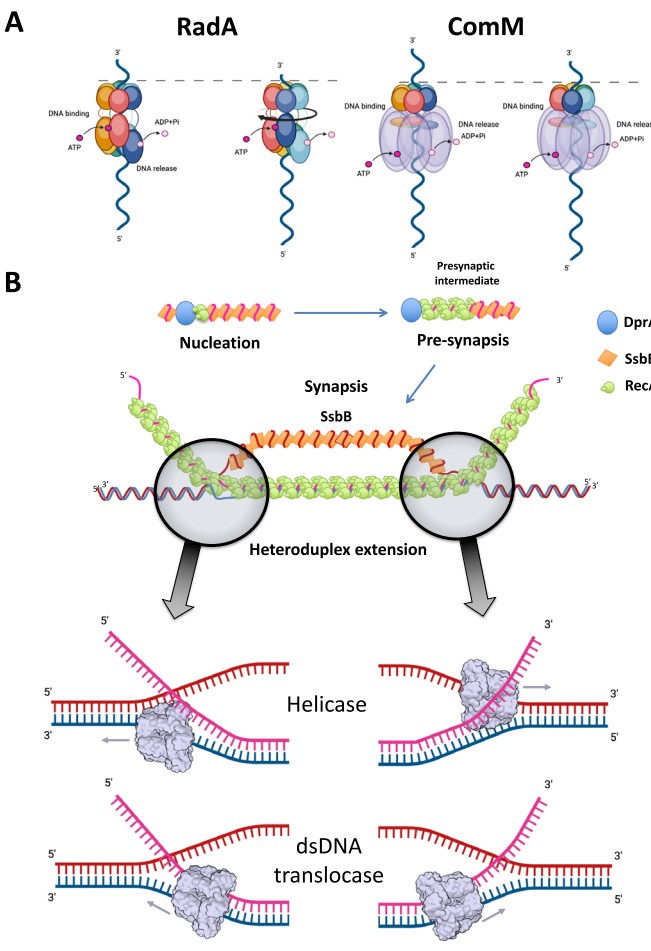

**Figure 6.** Mechanistic and functional models for RadA and ComM during homologous recombination.

(A) left panel, model for DNA translocation by RadA upon ATP hydrolysis. Rigt panel, model for DNA translocation by ComM upon ATP hydrolysis. (B) Potential roles for RadA or ComM during homologous recombination at the 3-strand junctions of the D-loop. Figure created using BioRender.com.

domain remain almost planar with respect to the Lon domain (Fig. 2B). In the related MCM proteins, this is achieved by two long flexible loops, named Ps1β and H2i, which form a continuous hydrophobic chamber for the translocation of dsDNA on the hexamer (Fig. 2) (Fernandez and Berger, 2021; Fletcher et al, 2003). Both loops are present in ComM, with the CL1 loop corresponding to Ps1β being very similar, but H2i being much shorter. The H2i region of MCM typically contains an aromatic residue, i.e. Y386 in archaeal MCM or W569 in human MCM2, which intercalates between DNA bases. This interaction has been proposed to either disrupt DNA stacking during strand separation or to provide a torque for translocation movement (Fletcher et al, 2003; Rzechorzek et al, 2020). The ComM H2i hairpin lacks an equivalent aromatic residue, but this may be compensated for by additional interactions between DNA and the Lon domain that are absent in MCM homologs. Indeed, the Lon domain of ComM contacts DNA in at least three positions: K43, K46 and K64 (Fig. 4E). In contrast to RadA, disruption of the Walker A motif in ComM results in a loss of DNA binding and helicase activity (Nero et al, 2018). Based

on the topological arrangement of the ATP-binding site relative to the DNA-binding loops, it is hypothesized that R316, which belongs to the sensor 3 motif, advances the CL1/Ps1β hairpin upon ATP binding. Similarly, E302, located in the Walker B motif, may have a similar effect on the H2i hairpin, and R381 may have a similar sensing function affecting the binding of the 366–373 region. This results in cooperative binding between DNA and ATP, as observed for RadA (Fig. 3B,D). Disruption of ATP cleavage could lead to conformational changes in the three sensor residues, resulting in the release of DNA (Fig. 6A).

## RadA and ComM as potential dsDNA translocases

Our structural analysis of the interaction of RadA and ComM with DNA stabilized by poorly or non-hydrolyzable ATP analogs provides insight into how they might promote DNA branch migration during HR. Previous biochemical analyses of RadA proteins from different organisms have reported common but also some important functional differences in their DNA binding and helicase activities with respect to ATP binding and hydrolysis (Cooper and Lovett, 2016; Inoue et al, 2017; Marie et al, 2017; Torres et al, 2019). They were found to be ATPases activated by either ssDNA or dsDNA. However, they differ in their ability to promote helicase activity in vitro, which was observed for *S. pneumoniae* and *B. subtilis* RadA, but not for *E. coli* RadA (and not reported for *T. thermophilus* RadA). However, *E. coli* RadA, like *S. pneumoniae* and *B. subtilis*, promotes 3-strand DNA branch migration in vitro (Cooper and Lovett, 2016). Remarkably, 4-strand DNA branch migration has also been tested and observed for *E. coli* RadA (Cooper and Lovett, 2016). Our structural and functional analysis of *S. pneumoniae* RadA highlighted its homology with DnaB helicases and suggested that it could promote 3-strand DNA branch migration by translocating onto ssDNA at the D-loop boundaries in the 5' to 3' direction, coupling dsDNA strand separation with ssDNA pairing (Marie et al, 2017). Applying such a reaction at each D-loop boundary would favor ssDNA recombination in both directions, as supported by the genetic analysis of the role of RadA in NT (Marie et al, 2017). However, DnaB helicases are also known to translocate along dsDNA and to promote DNA branch migration by translocating along duplex DNA (Kaplan and O'Donnell, 2002; Singleton et al, 2007). Such dsDNA encircling is exactly what was found in this structural analysis of RadA, which appeared to be stabilized in a translocation state with canonical DNA-binding loops interacting with one DNA strand (Fig. 3).

Interestingly, biochemical analysis of ComM from *V. cholerae* showed that it exhibited bidirectional helicase and DNA branch migration activities (Nero et al, 2018). Similarly, the cryoEM structure of ComM from *L. pneumophila*, which we have characterized here, revealed its interaction with dsDNA.

These findings support a model in which these two hexameric DNA motors act as dsDNA translocases to direct 3-strand or 4-strand DNA branch migration. In this model of mechanism, RadA would be functionally similar to RuvB and RecG (Fernandez and Berger, 2021). This would explain the ATPase activity of RadA, which can be activated by either ssDNA or dsDNA, as well as why *E. coli* RadA does not exhibit bona fide helicase activity while being able to branch migrate double-stranded branched DNA substrate. This model raises two important questions. First, how could the

Lon ring of RadA accommodate dsDNA? The average diameter of its positively charged pore is about 18 Å and is closed by flexible loops (residues 313–319) that define a constriction of 14 Å in diameter. B-form DNA has an average diameter of 18 Å. Therefore, in the conformation we observed, the inner channel of the RadA Lon domain should be opened to allow the passage of dsDNA, possibly by pushing away these flexible loops. In contrast, we show that ComM has a larger pore radius with a diameter of 24 Å. This diameter is similar to the inner channels found in dsDNA translocases such as FtsK (Jean et al, 2020). Interestingly, the densities corresponding to DNA in this region are quite weak in ComM, suggesting a higher degree of flexibility and/or conformational heterogeneity for the DNA. A second question raised by this model is how these two DNA motors are recruited to the HR intermediates to promote DNA branch migration by translocating on dsDNA. Some clues are provided by their common key role in the extension of ssDNA incorporation from the HR D-loop of NT. This intermediate could not evolve in any other branched DNA structure because one of the two exchanged DNA molecules is the invading linear ssDNA. Thus, to extend its recombination by translocating on dsDNA, these DNA motors could act by translocating the D-loop inwards on the duplex DNA assembled by RecA (Fig. 6B). At the three-way DNA junction, the protein would couple dsDNA unwinding and ssDNA pairing to extend recombination. Thus, RecA bound to dsDNA could define a landing pad for recruiting RadA and ComM inside the D-loop, giving them access to DNA for their translocation. In support of this model, physical and/or synergistic effects have been reported between RecA and RadA from *E. coli*, *S. pneumoniae*, and *B. subtilis* (Cooper and Lovett, 2016; Marie et al, 2017; Torres et al, 2019).

In conclusion, our structural and functional analysis of DNA-bound RadA and ComM provides important insights into how they act as DNA branch migration effectors during HR. These findings, combined with the previous biochemical analysis of these two hexameric motors that are widely conserved in bacteria, pave the way for future studies to uncover how they are recruited, loaded, and controlled in the HR pathways in which they specifically act, and how they mediate DNA strand exchange during their translocation.

## Methods

### RadA purification for CryoEM analysis

Full *radA* gene was amplified from *S. pneumoniae* genome and cloned in pASK-IBA3 vector, which adds a N-terminal strep-tag to the final protein product. After verification through sequencing, pASK-IBA3-*radA* was transformed into *E. coli* BL21 (DE3) for expression. The transformed strain was grown in $2 \times 2$ L LB media in 5 L conical flasks at 37 °C and 200 rpm until $OD_{595}$ of 0.8, induced with 200 μg/L of anhydrotetracycline and incubated for 20 h at 16 °C and 200 rpm. Cells were pelleted by centrifugation at $15{,}000 \times g$ for 20 min at 4 °C and resuspended in 50 ml of lysis buffer (Hepes 50 mM pH 8.0; NaCl 100 mM; $MgCl_2$ 2 mM) plus DNAse (5 μg/ml), Lysozyme 1 mg/ml and 1 tablet of EDTA-free protease inhibitor (ROCHE). Cells were broken passing twice through an Emulsiflex C5 (ATA scientific) and debris were pelleted by centrifugation at $100{,}000 \times g$ for 20 min. The supernatant was

loaded on a 5 ml Strep-tag purification column, washed with assay buffer (Hepes 50 mm pH 8.0; NaCl 100 mM) and eluted with 30 ml of assay buffer supplemented with 0.5 mg/ml Desthiobiotin (Sigma). Fraction-containing RadA were pulled together and injected in a 5 ml HiTrap heparin Column (GE) for DNA dissociation and eluted in assay buffer with a NaCl gradient ranging from 0.1 to 2 M. Protein presence was confirmed by electrophoresis in a 15% SDS-PAGE gel, run at 200 V for 40 min.

### ComM purification for CryoEM analysis

Full *comM* was amplified from *Legionella pneumophila* genome and cloned in pCDF-Duet1 to generate a C-terminal His-tagged ComM as a product. After verification by sequencing, pCDF-Duet1-*comM* was transformed in BL21 (DE3) for expression. The transformed strain was grown in $2 \times 2$ L LB media in 5 L conical flasks at 37 °C and 200 rpm until $OD_{595}$ of 0.8, induced with 1 mM IPTG and incubated for 20 h at 16 °C and 200 rpm. Cells were recovered, lysed and centrifuged as for RadA. The supernatant was loaded on a 5 ml His-trap column (GE), washed with assay buffer supplemented by 20 mM Imidazole and eluted with assay buffer supplemented with 500 mM imidazole. Fractions containing ComM were pulled together, concentrated to 0.5 ml using a Vivaspin column with 30 kDa MWCO and injected in a Superdex-200 gel filtration column, run with assay buffer. ComM eluted as a single peak at 13 ml, consistent with a molecular weight of 50 kDa. Protein presence was confirmed by electrophoresis as for RadA.

### CryoEM sample preparation

Synthetic oligos were ordered from Eurofins. Hybrid DNA was prepared by mixing semi-complementary oligonucleotides (Appendix Table S1) at 100 μM final concentration of each in assay buffer supplemented with 1 mM DTT, boiled for 5 min and letting it cool down at room temperature. DNA-binding reaction for RadA was started by mixing 7 μM RadA (0.35 mg/ml), 10 μM of hybrid DNA (Appendix Table S1, combination CD) and 5 mM ATP-γ-S, followed by a 30 min incubation at 37 °C. DNA-binding reaction for ComM was started by mixing 18 μM ComM (1 mg/ml), 10 μM of hybrid DNA (Appendix Table S1, combination EG) and 5 mM AMPPNP, followed by a 10 min incubation at 37 °C.

Quantifoil Cu300 mesh R2/2 grids were glow discharged at 0.3 mbar vacuum and 2 mA current for 35 s. 4 μl of reaction was added to the grid and blotted for 2.5 s in a Vitrobot instrument (FEI), at 100% humidity and 4 °C, at blot force 0, before being plunge-frozen in liquid ethane and transferred/stored in liquid nitrogen.

### RadA CryoEM data collection and processing

Sample screening and pilot data collection was performed in a 200 kV Talos Arctica equipped with a K2 summit camera at the Institut Européen de Chimie et Biologie (Pessac-FR) using SerialEM (version 3.8) software. Final RadA Data collection was performed in a 300 kV Titan Kryos cryo-microscope (EFI) at ESRF (Grenoble-FR) (experiment MX-2261) at 165k magnification, pixel size of 0.827 Å, total dose of 54 e⁻/Å² and 4.5 s exposition, using EPU software. A total of 8983 movies were collected and processed using Relion 3.1 software (Scheres, 2012). In total, 2000 particles were manually picked and submitted to 2D classification in order to

generate initial templates, which were in turn used for template-based picking. In total, 908,107 particles were extracted and submitted to three rounds of 2D classification, resulting in 263,835 selected particles. The generated 2D classes were used as templates for an iterative round of template-based picking, which resulted in the picking and extraction of 3,768,530 particles. After several rounds of 2D classification, 1,258,631 particles were selected for a consensus 3D refinement without symmetry imposition, which generated a 3.54 Å overall resolution map, but in which only the Lon protease domains were well-defined. This consensus map was used as a template for 3D classification without image alignment with a four-class separation in order to further clean bad particles. In all, 1,208,572 particles from good classes were selected, and submitted to a new consensus refinement, followed by a new 3D classification without image alignment in a 10-classes separation. The best class, containing 303,651 particles, was selected and refined, generating a 3.6 Å overall resolution map. After per-particle CTF correction and Bayesian polishing, a final refinement was performed, generating a 3.15 Å overall resolution map. A scheme of the data processing is presented in Appendix Fig. S7.

The final map was post-processed using Autosharpen tool from Phenix package (Liebschner et al, 2019) using the two half maps in the input. Separated domains extracted from RadA's crystallographic structure (Marie et al, 2017) were manually fit in the map using Chimera software, followed by a rigid body fitting using the RealSpaceRefine tool from the Phenix package (Liebschner et al, 2019). Manual curation and building of missing regions, as well as DNA and ATP-γ-S fitting, was performed in *Coot*, followed by another round of RealSpaceRefine. Validation was first performed in *Coot* (Emsley et al, 2010) and subsequently checked using 'CryoEM comprehensive validation' tool from Phenix package (Liebschner et al, 2019). Statistics for image processing and structure modeling are available in Appendix Table S3.

## ComM cryoEM data collection and processing

Sample screening and pilot data collection were performed as for RadA. Final ComM data collection was performed in a 300 kV Titan Kryos cryo-microscope (EFI) equipped with a Quantum-K3 camera at EMBL (Heidelberg, Germany) at 130k magnification, pixel size of 0.645 Å, total dose of 53.56 e$^-$/Å$^2$ and 1.2 s exposure, using SerialEM Version 3.8. A total of 32,534 movies were collected and processed in parallel using Cryosparc 3.3.1 and Relion 4.0 software (Punjani et al, 2017; Scheres, 2012). In Cryosparc, Movies were processed using PatchMotionCor and Patch CTF estimation. A subset of 2000 micrographs was used for blob picking with particle diameter of 100 Å. After 2D classification, 32,000 particles were selected and used as an input for picking using Topaz (Bepler et al, 2019). After the removal of duplicates, 1,687,279 particles resulted from Topaz picking, which were subjected to 2D classification, where 852,356 particles were selected. Data processing was twofold. In the first approach, the selected particles were subjected to ab-initial model generation with 10 classes. All the resulting models and particles were used as an input for a 10-class heterogeneous refinement. The promising classes were pulled together and submitted to a new round of ab-initial model generation, with 6 classes, and again, all the models were submitted

to heterogeneous refinement with 6 classes. Two of these classes were selected, and submitted to unmasked refinement, followed by masked local refinement on the DNA-bound hexamer. This generated two subpopulations of DNA-bound ComM maps (main map EMDB 19574 PDB 8RXD, and subpopulation 2, EMDB 19575). In the second, classical approach, selected particles were subjected to ab-initial reconstruction with three classes. The best resulting model was used for a homogeneous refinement, resulting in an unmasked map of 3.17 Å, with well-defined Lon domain regions, but showing high variability on the ATPase region. This map was used as an input for a Non-Uniform refinement with a mask on the ComM hexamer, resulting in a hexamer consensus map of 3 Å (EMDB 19581). This consensus map was used as an input for a focused refinement on the Lon region, resulting in a 2.8 Å map for this region (EMDB 19578 PDB 8RXS). The hexamer consensus map was subjected to hexamer-masked hetero-refinement with 4 classes. The selected class contained 305,846 particles, and was subjected to a local refinement using a mask on half of the hexamer (a trimer) + DNA, resulting in a local map of 3.23 Å with considerably more homogeneity in the ATPase region (EMDB 19577 PDB 8RXK). A scheme of the data processing is shown in Appendix Fig. S7.

The initial molecular model for the ComM monomer was generated using Alphafold2 software (Jumper et al, 2021) The resulting model was divided into two domains (residues 1–180 and 181–500), copied 6 times to form a hexamer and manually fit in the main map using UFSC chimeraX (Meng et al, 2023). A 24mer dsDNA was generated in Chimera and fitted likewise. This model was submitted to initial adjustments in ISOLDE (Croll, 2018), followed by refinement in PHENIX RealSpaceRefine. Further adjustments were performed in *Coot* and ISOLDE before a final round of refinement and validation in PHENIX.

For the dodecamers, the 852,356 particles from the consensus map were submitted to 3D classification with two classes, one of which showed initial density for ComM dodecamerisation, with 464,656 particles, but the second dodecamers had poor density due to high heterogeneity. After an unmasked 3D refinement, the subset was subjected to hetero-refinement with 10 classes. The best class, containing 69,029 particles, was refined, resulting in a 4.1 Å map of ComM dodecamers (EMDB 19580—Appendix Fig. S7). Two ComM hexamer models were fit in the dodecamer density using UFSC Chimera, excluding the DNA chains. A 48mer dsDNA was generated in Chimera and fitted in the density, and further adjusted using ISOLDE.

For the DNA-free ComM hexamer, in Relion 4.0, Movies were processed using MotionCor2 (Zheng et al, 2017), with 10 × 10 patches, and Gctf (Zhang, 2016). Laplacian picking was used on a 1000 micrographs subset. After classification, 10,000 particles were used as input for Topaz picking on the whole dataset (Bepler et al, 2019), resulting in 2,888,884 picked particles. After rounds of 2D classification, 621,282 particles were selected and submitted to initial model generation. The initial model was refined without mask, resulting in a 6 Å consensus model. 3D classification with 10 classes and no particle alignment was performed, which revealed a class containing 78,222 particles of DNA-free ComM hexamers. This class was subjected to masked refinement, generating a 4.6 Å map. After CTF correction, Bayesian polishing, and a final masked refinement resulted in a 3.93 Å map of DNA-free ComM hexamers (EMDB 19579 PDB 8RXT—Appendix Fig. S7).

Statistics for image processing and structure modeling are available in Appendix Table S3.

## RadA production and purification for biochemical assays

The plasmid IBA13 (Amp^R, IBA Lifesciences) was previously modified to harbor the gene encoding the superfolder Green Fluorescent Protein (sp-GFP) in frame with the Strep-tag®II encoding sequence followed by a TEV cleavage site under the Tet promoter. The gene *radA* was cloned from a pneumococcal R800 derivative strain into the modified IBA13 plasmid by FastCloning (Li et al, 2011), downstream of the TEV cleavage site. This plasmid allows the production of the GFP-RadA fusion protein with a Strep-tag®II at the N-terminus and a TEV cleavage site for the removal of the GFP and the tag.

The RadAR301A mutant was generated by site-directed mutagenesis PCR on the plasmid using a single primer (EVo-21, see sequence in Table 2), the PfuTurbo DNA polymerase (Agilent Technologies) and the Taq DNA Ligase (New England Biolabs) according to a protocol adapted from (Shenoy and Visweswariah, 2003).

*E. coli* BL21-Rosetta™(DE3) competent cells (Cm^R, Novagen) were transformed with the expression vector by heat shock. Transformants were grown at 37 °C in Terrific Broth medium complemented with 100 µg/mL Ampicillin (Amp) and 10 µg/mL Chloramphenicol (Cm) to an optical density of about 0.6. The culture was cooled down for 1 h at 16 °C, then the protein expression was induced by adding 200 ng/mL Anhydrotetracycline and incubating at 16 °C overnight. Cells were harvested by centrifugation and resuspended in Sucrose Buffer (25 mM Tris pH 8.0, 25% sucrose, 1 mM EDTA) complemented with protease inhibitor (cOmplete™, EDTA-free, Sigma-Aldrich) and 200 µg/mL lysozyme and flash-frozen in liquid nitrogen to be stored at −80 °C. Cells were thawed at room temperature and lysed by sonication. Cell debris was pelleted, and clear lysate was loaded on StrepTactin® resin (IBA Lifesciences) pre-equilibrated with Strep Buffer (50 mM HEPES pH 8.0, 300 mM NaCl, 2 mM MgCl$_2$). After a wash step with Strep Buffer, GFP-RadA was eluted with Strep Buffer complemented with 2.5 mM desthiobiotin (IBA Lifesciences). The fusion protein was cleaved with ~80 µg/mL TEV protease (homemade) at 10 °C overnight. Cleaved GFP and desthiobiotin were removed by anion exchange chromatography on a HiTrap Q HP column (GE Healthcare) connected to a FPLC system (ÄKTA purifier-10, GE Healthcare). The cleaved sample

was diluted to 100 mM NaCl and loaded on the column pre-equilibrated with Buffer A (50 mM HEPES pH 8.0, 100 mM NaCl, 2 mM MgCl$_2$). The column was washed with Buffer A, and the proteins were eluted with a linear gradient of 0.1–1 M NaCl. RadA was then separated from uncleaved GFP-RadA on StrepTactin® and concentrated by anion exchange chromatography. The final fractions of RadA (containing about 300 mM NaCl) were stored at −20 °C after the addition of 50% glycerol or at −80 °C after the addition of 10% glycerol and freezing in liquid nitrogen.

## DNA substrates for DNA-binding, ATPase, and helicase assays

Labeled and unlabeled oligonucleotides were purchased from Eurogentec and Eurofins Genomics, respectively.

Single-stranded DNA (ssDNA) for DNA-binding assays is a 100 nt-long oligonucleotide labeled at its 3' end with the Cy3 dye (Ovio9, see sequence in Table 1). Double-stranded DNA (dsDNA) for DNA-binding assays is the 3'Cy3-labeled 100 bp oligonucleotide hybridized to a fully complementary unlabeled 100 nt oligonucleotide (Olea41).

ssDNA for ATPase assays is a 60 nt-long oligonucleotide (Ovio45, see sequence in Table 1).

The hybrid DNA substrate for helicase assays is a 30 nt-long oligonucleotide labeled at its 5' end with the Alexa Fluor 532 (AF532) dye (OA532nt30, see the sequence in Table 1) hybridized to a 100 nt-long oligonucleotide labeled at its 3' end with the Black Hole Quencher™ BHQ-1 (oligoBHQ100).

Hybrid and dsDNA substrates for helicase and DNA-binding assays respectively were generated by mixing the two strands at 5 µM each (except oligoBHQ100 at 6 µM, slight excess) in 20 mM Tris pH 7.5, 100 NaCl, 10 mM MgCl$_2$, 1 mM dithiothreitol (DTT), incubating 5 min at 100 °C, then placing in boiling water and allowed to cool down to room temperature.

## DNA-binding assays

100, 300, and 900 nM RadA was incubated at room temperature for 15–20 min with 10 nM DNA, 10 mM magnesium acetate, and 6% glycerol in a 10 µL reaction solution containing 50 mM Tris pH 7.5, 100 M NaCl, 1 mM DTT, 0.1 mg/mL bovine serum albumin (BSA). Samples were loaded on a 4% acrylamide gel (acrylamide:bisacrylamide 29:1, 0.3× TBE) and migrated in 0.3× TBE for 30–70 min at 200 V. The gel was visualized using a Typhoon 9400 (GE Healthcare)

**Table 1.  Oligonucleotides used for activity tests.**

| Oligonucleotide (nt) | Sequence (5' to 3') | Used in this work |
|---|---|---|
| Ovio9 (100) | AATCATGGTCATAGCTGTTTCCTGTGTGAATAGCAATGTAATCGTCTATGACGTTAA ACCATCGATAGCAGCACCGTAATCAGTAGCGACAGAATCAAGT-**Cy3** | DNA-binding assay |
| Olea41 (100) | ACTTGATTCTGTCGCTACTGATTACGGTGCTGCTATCGATGGTTTAACGTCATAGACGATTACATTGCTA TTCACACAGGAAACAGCTATGACCATGATT | DNA-binding assay |
| Ovio45 (60) | CTAGGGTCGGATCCTCTAGACAGCTCCATGATCACTGGCACTGGTAGAATTCGGCCCATT | ATPase assay |
| OA532nt30 (30) | **AF532**-CTAGGGTCGGATCCTCTAGACAGCTCCATG | Helicase assay |
| oligoBHQ100 (100) | ACTTGATTCTGTCGCTACTGATTACGGTGCTGCTATCGATGGTTTAACGTCATAGACGATTACATTGCTA CATGGAGCTGTCTAGAGGATCCGACCCTAG-**BHQ-1** | Helicase assay |
| Oas30nt (30) | CATGGAGCTGTCTAGAGGATCCGACCCTAG | Helicase assay |

The modifications (fluorophore or quencher) are indicated in bold.

**Table 2. Oligonucleotides used for pneumococcal strain mutagenesis.**

| Primer name (nt) | Sequence (5'–3') | Comments |
|---|---|---|
| oALS20 (23) | GGTTGTGTGATTGTCAAAGATGG | Forward primer, 1638 bp upstream *radA* initiating codon |
| EVo-21 (47) | CGTTGTAACCATGGAAGGGACG**GC**TCCGATTTTGGCGGAGGTTCAGG | Mutation forward primer |

with excitation/emission wavelengths of 532/580 nm. The intensity of free DNA bands was quantified using MultiGauge V3.0 (Fujifilm).

## ATPase assays

The ATPase activity of RadA was determined using an ATP regeneration system coupled to NADH oxidation (Kiianitsa et al, 2003; Sehgal et al, 2016). Overall, 300 nM RadA was mixed with 1.2 mM phospho(enol)pyruvic acid (Sigma-Aldrich), 70 to 120 U/mL pyruvate kinase (Sigma-Aldrich), 200 U/mL L-lactic dehydrogenase (Sigma-Aldrich), 0.25 mg/mL NADH (Sigma-Aldrich), 2 $\mu M_{nt}$ ssDNA (Ovio45) and 2 mM ATP (GE Healthcare) in a 40 µL reaction solution containing 10 mM Tris-HCl pH 7.5, 4 mM magnesium acetate, 0.1 mM DTT. The reaction was carried out in a 96-well plate at 25, 37, or 42 °C on a VarioSkan Flash microplate reader (Thermo Scientific). The NADH absorbance was measured at 340 nm and data were analyzed using Excel (Microsoft). The absorbance decrease was converted to the amount of NADH oxidized (equal to the amount of ATP hydrolyzed) using the NADH extinction coefficient at 340 nm of 6220 $M^{-1}.cm^{-1}$.

## Helicase assays

The helicase activity of RadA was determined by measuring the fluorescence intensity increase resulting from the DNA substrate unwinding (separation of fluorescent OA532nt30 and quencher oligoBHQ100 strands). 300 nM RadA was mixed with 10 nM DNA substrate, 100 nM capture strand (Oas30nt, fully complementary to the fluorescent strand to avoid it re-hybridizing to the quencher strand), and 5 mM ATP (GE Healthcare) in a 100 µL reaction solution containing 10 mM Tris pH 7.5, 50 mM NaCl, 10 mM magnesium acetate, 1 mM DTT, 0.1 mg/mL BSA. The reaction was carried out in a quartz cuvette on a FluoroMax spectrofluorimeter (HORIBA) equipped with a circulating water bath set to 37 °C. The fluorescence was measured at 532/554 nm (excitation/emission).

## Pneumococcal radA mutagenesis

To generate strain R4735 (with the R301A mutation in RadA), PCR fragments of regions upstream and downstream of the *radA* gene were amplified, with 2 nt of *radA* mutated using primer pairs oALS20-oALS30 and Evo-21-oALS34 (see sequences in Table 2). Splicing overlap extension (SOE) PCR with these two fragments as templates and the primer pair oALS20-oALS34 was carried out to generate a PCR fragment with the *radA* gene mutated and ~2.5 kb of homologous sequence on either side. This DNA fragment was transformed into strain R1818 (WT, Caymaris et al, 2010) without selection. Briefly, 100 µL aliquots of pre-competent cells were resuspended in 900 µL fresh C + Y medium (Martin et al, 1995) with 100 ng mL$^{-1}$ competence stimulating peptide (CSP) and incubated at 37 °C for 10 min. Transforming DNA was then added

to a 100 µL aliquot of this culture, followed by incubation at 30 °C for 20 min. Cells were then diluted in 1.4 mL C + Y medium and incubated at 37 °C for 3 h 30 min, before dilution and plating without selection. Transformants were identified by sequencing.

## Transformation assays

To test the efficiency of transformation in wild-type (WT) and mutant strains, 100 µL pre-competent cultures were resuspended in 900 µL fresh C + Y medium pH 7.6, and 100 ng mL$^{-1}$ CSP was added. Cells were incubated for 10 min at 37 °C, before 100 µL was added to tubes containing desired transforming DNA. Transforming DNA was a 4.2 kb-long fragment carrying the *rpsL41* Sm$^R$ allele amplified from R304 chromosomal DNA using the RpsL5-RpsL6 primer pair (Bergé et al, 2013) and providing resistance to Streptomycin. Transforming cultures were incubated at 30 °C for 20 min before dilution and plating of appropriate dilutions in 10 mL CAT agar medium (Porter and Guild, 1976; Bergé et al, 2013) with 3% horse blood. Plates were incubated 2 h at 37 °C, before addition of a second 10-mL layer of CAT agar medium with or without Streptomycin for transforming and total cells respectively. Plates were incubated overnight at 37 °C and colonies present on selective and non-selective plates were compared to calculate transformation efficiency.

## Data availability

RadA model information is available under accession PDB 8RXC and the density map under accession EMDB 19573. dsDNA-bound ComM main map is available under EMDB 19574 and PDB 8RXD, while the map relating to subpopulation 2 is under EMDB 19575. Coordinates of ComM Lon domain are available under the accession codes PDB: 8RXS and EMDB 19578. Coordinates for dsDNA-bound ComM trimer are available under accession code PDB: 8RXK and EMDB 19577. Density map for the dsDNA-bound ComM hexamer used for focused refinement are submitted under coordinates EMDB 19581. Coordinates and density map for ComM hexamer in the absence of DNA are available under accession codes PDB: 8RXT and EMDB 19579. Density map for ComM dodecamers coordinating dsDNA area available under EMDB 19580 accession code.

The source data of this paper are collected in the following database record: biostudies:S-SCDT-10_1038-S44318-024-00264-5.

## Peer review information

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

## Acknowledgements

This project was executed with funds from the European Research Council Consolidator Grant TransfoPneumo (grant 725554). We acknowledge the European Synchrotron Radiation Facility for the provision of beam time on CM01 for collection on RadA, and we would like to thank Michael Hons for assistance. This work benefited from access to the CryoEM platform of EMBL Heidelberg and has been supported by iNEXT-Discovery, project number 871037, funded by the Horizon 2020 program of the European Commission, with assistance of Felix Weis. EV received a fellowship from the Fondation pour la Recherche Médicale (grant FRM SPF20170938718). We thank Xavier Charpentier for providing the genomic DNA of *L. pneumophila paris*.

## Author contributions

**Leonardo Talachia Rosa**: Conceptualization; Investigation; Methodology; Writing—original draft; Writing—review and editing. **Émeline Vernhes**: Conceptualization; Investigation; Methodology; Writing—original draft; Writing —review and editing. **Anne-Lise Soulet**: Methodology. **Patrice Polard**: Supervision; Funding acquisition; Investigation; Writing—original draft; Project administration; Writing—review and editing. **Rémi Fronzes**: Conceptualization; Formal analysis; Supervision; Funding acquisition; Writing—original draft; Project administration; Writing—review and editing.

Source data underlying figure panels in this paper may have individual authorship assigned. Where available, figure panel/source data authorship is listed in the following database record: biostudies:S-SCDT-10_1038-S44318-024-00264-5.

## Disclosure and competing interests statement

The authors declare no competing interests.

# Expanded View Figures

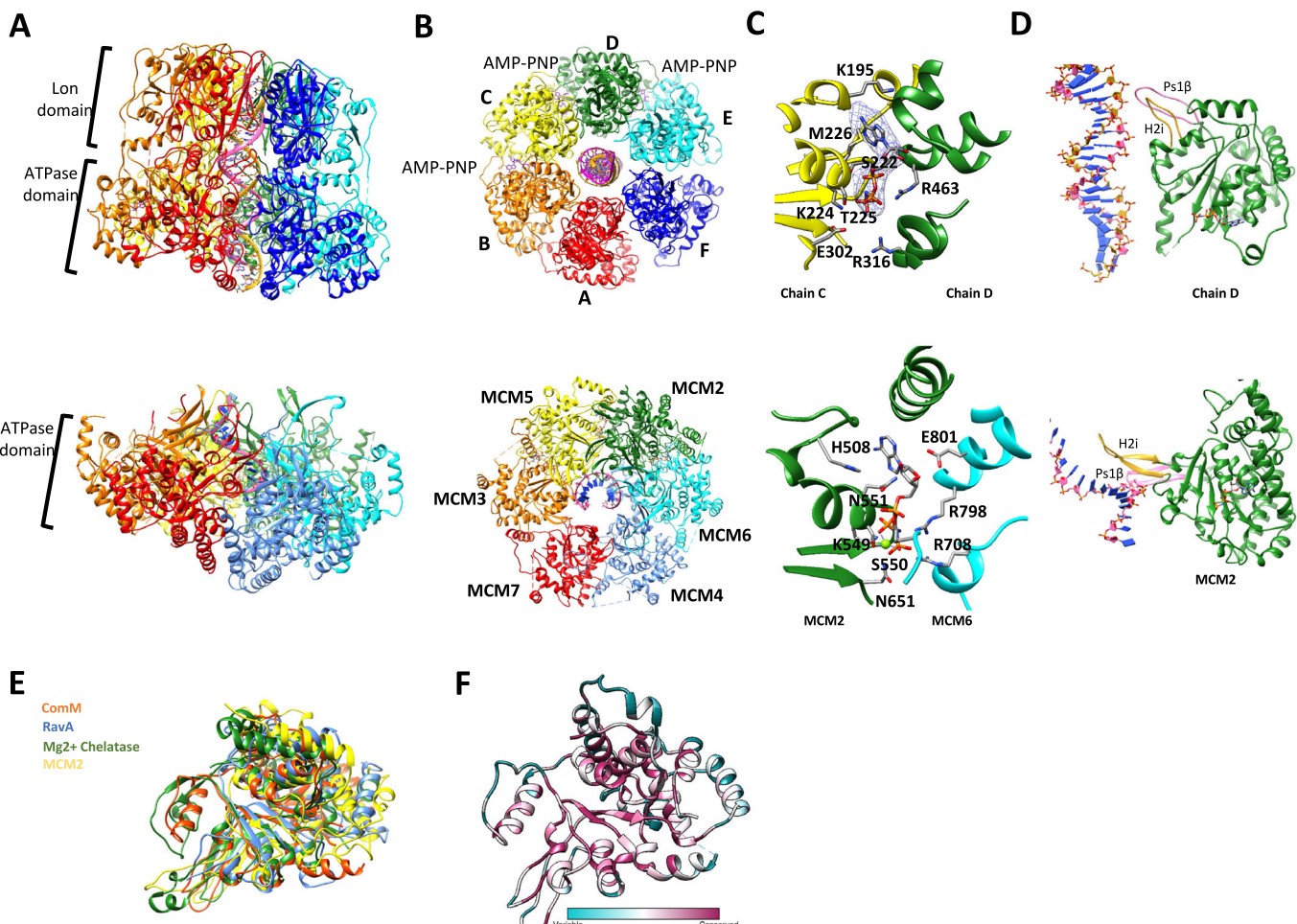

**Figure EV1.   Comparison between ComM and the C-terminal ATPase domains of human MCM (PDB: 6SKL), RavA (PDB 6SZB) and magnesium chelatase (PDB 1G8P).**

(A) Arrangement around the DNA for ComM (top) and MCM (bottom), with MCM colored according to ComM scheme, from MCM7 to MCM4. (B) View from the top on the N-terminal side of ComM (top) and MCM (bottom). (C) ATP coordination site in ComM (top) and MCM (bottom). In ComM, AMPPNP is depicted in sticks, colored by element, with the cryoEM density map depicted as a blue mesh. Chains C and D are depicted as yellow and green ribbons, respectively, with relevant residues depicted in sticks and colored by element. In MCM, MCM2 is depicted in green, while MCM6 is depicted in cyan. AMPPNP is depicted as sticks, colored by element, and so are relevant residues. (D) Comparison between DNA-binding loops of ATPase domains in ComM (top) and MCM (bottom). The ATPase domains of ComM monomer D and MCM2 are depicted in green ribbons, with ps1B loop depicted in pink and H2i in golden. (E) Superposition of ComM, RavA, $Mg^{2+}$-chelatase and MCM2. ComM ATPase domain (residues 190–500) is shown in orange, RavA ATPase domain is depicted in blue (PDB 6SZB, residues 3–306), and shows an RMSD of 2.176 Å with ComM. $Mg^{2+}$ Chelatase (PDB 1G8P) is shown in green, and has an RMSD of 1.928 Å with ComM. MCM2 ATPase domain is depicted in yellow (PDB 6RAW- chain 2, residues 447–798), and has a RMSD of 1.5 Å over 56 pruned atoms. (F) Consurf analysis of ComM ATPase domain (residues 190–500), showing conservation of residues along homologous sequences, depicted in a color gradient from cyan (variable) to purple (conserved).

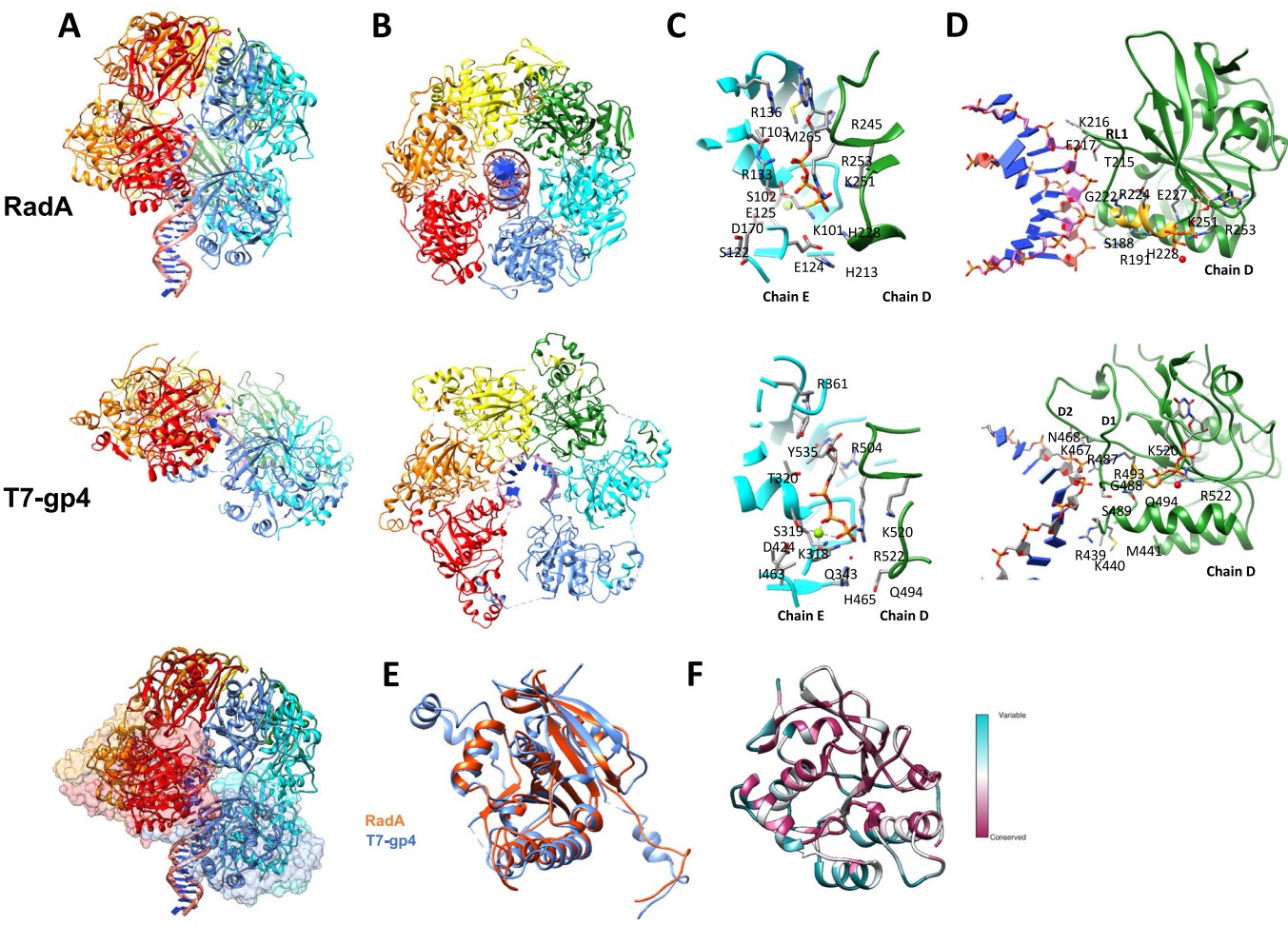

**Figure EV2. Comparison between RadA and the ATPase region of the superfamily 4 helicase T4-gp4 (PDB: 6N7I).**

(A) Overall structure of RadA (top) and T7-gp4 (middle). Both helicases present a helical arrangement, tilted in relation to the DNA helix. Bottom panel depicts RadA in ribbon representation, fit inside the transparent surface density of T7-gp4. (B) Top view from the C-terminus of RadA (top panel) and T7-gp4 (bottom panel). (C) ATP coordination sites of RadA (top) and T7-gp4 (bottom). ATP-γ-S (in RadA) and ATP (in T7-gp4) are depicted as sticks colored by element. Ribbon representation for Chain D is shown in green and chain E in cyan. Relevant residues are shown as sticks, colored by element. (D) Cooperative coordination of ATP and DNA in RadA (top) and T7-gp4 (bottom). ATP-γ-S (in RadA) and ATP (in T7-gp4), as well as relevant residues, are depicted in stick representation colored by elements. The ATPase domains are depicted as green ribbon, with the region structured upon ATP binding highlighted in golden. DNA is depicted as sticks, with the bases represented as blue rectangles. (E) Superposition of the ATPase domains of RadA (orange, residues 54–273) and T7-gp4 (blue, residues 263–547), resulting in an RMSD of 1.16 Å over 120 pruned residues or 1.944 Å overall. (F) ConSurf analysis of the ATPase domain of RadA, showing conservation of residues along homologous sequences, depicted in a color gradient from cyan (variable) to purple (conserved).

