## [Peer Review File · The EMBO Journal]

Structural insights into the mechanism of DNA branch migration during homologous recombination in bacteria

Remi Fronzes, Leonardo Rosa, Emeline Vernhes, Anne-lise Soulet, and Patrice Polard

Corresponding author(s): Remi Fronzes (r.fronzes@iecb.u-bordeaux.fr) , Patrice Polard (patrice.polard@univ-tlse3.fr)

Review Timeline:

Submission Date:	23rd Feb 24
Editorial Decision:	10th Apr 24
Revision Received:	9th Jul 24
Editorial Decision:	19th Aug 24
Revision Received:	6th Sep 24
Accepted:	20th Sep 24

Editor: Ieva Gailite

Transaction Report:

Dear Remi,

Thank you for submitting your manuscript for consideration by the EMBO Journal. I apologise for the protracted assessment of your manuscript due to delays in referee report submission and the high manuscript submission rate to our office at the moment. We have now received comments from three reviewers, which are included below for your information.

As you can see, particularly reviewers #1 and #3 find the study of interest, while all reviewers also indicate several aspects in which data presentation, interpretation and description should be improved in the revised manuscript. Based on the interest expressed by reviewers #1 and #3, I would like to invite you to address the comments of all reviewers in a revised version of the manuscript.

We generally allow three months as standard revision time, which can be extended if necessary. As a matter of policy, competing manuscripts published during this period will not negatively impact on our assessment of the conceptual advance presented by your study. However, please contact me as soon as possible upon publication of any related work to discuss the appropriate course of action. Should you foresee a problem in meeting this deadline, please let me know in advance to discuss an extension.

When preparing your letter of response to the referees' comments, please bear in mind that this will form part of the Review Process File and will therefore be available online to the community. For more details on our Transparent Editorial Process, please visit our website: <https://www.embopress.org/page/journal/14602075/authorguide#transparentprocess>. Please also see the attached instructions for further guidelines on preparation of the revised manuscript.

Please feel free to contact me if you have any further questions regarding the revision. Thank you for the opportunity to consider your work for publication. I look forward to receiving your revised manuscript.

With best wishes,

Ieva

Ieva Gailite, PhD
Senior Scientific Editor
The EMBO Journal
Meyershofstrasse 1
D-69117 Heidelberg
Tel: +4962218891309
i.gailite@embojournal.org

We realize that it is difficult to revise to a specific deadline. In the interest of protecting the conceptual advance provided by the work, we recommend a revision within 3 months (9th Jul 2024). Please discuss the revision progress ahead of this time with the editor if you require more time to complete the revisions. Use the link below to submit your revision:

Referee #1:

The paper by Talachia Rosa and colleagues reports the cryoEM structures of RadA and ComM, two ATPase motor proteins responsible for branch migration of D-loop boundaries during natural transformation in bacteria, together with accompanying biochemical characterisation of structure-based mutants. The novelty of the present work consists principally in the detailed elucidation of the mechanism of DNA binding by RadA and ComM. Many questions remain concerning the mechanism of branch migration by these proteins but in my view the presented evidence represents a sufficiently significant advance to warrant publication in EMBOJ.

Overall, the data presented by the authors is impressive for extent and quality. I don't have major concerns, but I would like the authors to rewrite the points below in the Discussion. The English could also be improved (see a couple of examples below).

Discussion

In general, the discussion of the mechanism of RadA loading onto dsDNA should be improved. The authors should highlight that it is highly unusual for a ring helicase/translocase to self load onto DNA, like they seem to imply. Both the MCM and DnaB helicases are loaded onto origin DNA by specific protein loaders. What are the mechanistic steps that would lead from DNA-free RadA to DNA-bound RadA ring, such as seen by the authors in their cryoEM work?

Related to this, the MCM helicase does not self-load onto origin DNA (line 420)! Surely the authors are aware that the Origin Recognition Complex is responsible for loading the MCM ring onto origin DNA.

Again related to this, the authors should also mention that the use of relatively short linear DNA substrates might have caused RadA to be threaded onto the DNA, rather than self-loading in a manner that is reflective of its true mechanism.

The observation that ComM can form double-hexamers on dsDNA is intriguing. Is it known whether ComM can branch migrate independently from RadA? And how do the authors envisage that RadA and ComM might cooperate in branch migration?

Methods

1. Show SDS-PAGE of purified RadA and ComM.
2. "CryoEM sample preparation" - give protein concentration in molarity, not mg/ml.

Minor points

The authors occasionally refer in the text to the 'lagging strand' (for instance on line 270: "A second loop named RL2, which consists of residues 170 to 189, coordinates the lagging 3' to 5' strand."). It is not clear what they mean by lagging strand outside of DNA replication. Please clarify.

The English could be made more accurate to improve the readability of the manuscript. For instance, consider the sentence on line 185: "Notably, although ComM and RadA have an inverted sequence order, they share identical topology between the ATPase and Lon domains". It's not the sequence, it's the arrangement of ATPase and Lon domains that is swapped. Also the structures show that they have identical reciprocal arrangements of ATPase and Lon domains, not identical topology. Please correct.

On line 166, "We overexpressed ComM from *L. pneumophila* in *E. coli*, attaching a C-terminal His6-tag." Please use 'fused to' rather than 'attaching'.

Please write "AMPP-N-P" as AMPPNP.

Referee #2:

RadA and ComM are important bacterial helicases involved in natural transformation with branch migration activity. In this manuscript, Rosa et al. report the structures of RadA and ComM in complex with their DNA substrates. These results would contribute to the understanding of the functions of RadA and ComM in branch migration. With this said, however, this study provides little insight into the mechanisms of these two helicases.

Moreover, the current version of the manuscript does not meet the criteria for being considered for acceptance, and should be largely revised. Most of the manuscript is on structural descriptions, which are too lengthy and hard to follow. Some descriptions appear inappropriate or lack support from structural information. The figures were inadequately prepared and did not effectively support the structural descriptions. Please refer to previously published structural papers in the EMBO J to improve the manuscript and figures.

Here are some examples for improvement:

- (1) Line 146-147, the flexibility of subunit F was not shown in Figure S1.
- (2) Line 151-153, where is the N-terminal arm in Figure 2a? All the domains, motifs or residues mentioned in the text should be clearly labeled in the supporting figures.
- (3) Line 153-154, where is the ZF domain in Figure S1.
- (4) Line 161-163, what does "DNA-B" mean? The B form DNA? The descriptions of some concepts or definitions should be accurate. Please show the structural evidence for the "distorted or disrupted" downstream DNA region. In addition, when the authors assumed that the downstream DNA around the Lon domain is distorted or disrupted, why did they propose that RadA captured the dsDNA end (line 160-161)?
- (5) Line 168-169, where is the 50 kDa monomer protein in Figure S3.
- (6) Line 180-181, the sentence discusses the domain similarity between the AAA+ ATPase domain of ComM and MCM. It raises the question as to why the authors did not present a superimposition of the ComM domain with that of MCM, similar to what was done for the domain comparison of ComM, RavA, and Chelatase.
- (7) Line 184, where is the ZF domain?
- (8) Line 187, label all the mentioned residues in figures.
- (9) Line 191, label the chelatase domains.
- (10) In lines 256-259, the description lacks sufficient evidence. Please provide a detailed explanation of the molecular mechanism by which histidine stacks with the short helix and how the formation of the helix pushes forward the RL1 loop. Otherwise, consider removing this sentence.
- (11) Line 295-296, as the interaction between Lon domain and DNA was not observed in the structure, is there direct evidence to support this description?
- (12) Line 269-273, where are the RL2 and residues 170-189?
- (13) Line 277-278, this description lacks a supporting figure.
- (14) Line 287-289, please show these interactions in detail.
- (15) Line 290-292, please present the "half size" in figures.
- (16) Line 293-294, figure 3f didn't support this conclusion.
- (17) Line 298, what does the lagging strand mean?
- (18) Line 327, the characterization of the R301-mediated domain interaction as a molecular switch is not justified. While the interaction of R301 in the Lon domain with E239/E76 is crucial for inter-domain interaction, it likely represents only a fraction of the intricate interactions between the Lon and ATPase domains. It is an exaggeration to claim R301 interaction as a molecular switch. The other residues contributing to the interface should also be described.
- (19) Line 339-340, what does this sentence mean?
- (20) Line 226: G100 could not be found in figure 3a.
- (21) Line 234: the PDB 6MII cannot be found in S5.
- (22) In figure 2c, ATPyS should be ATPγS. The display mode is not very clear in figure 2.

Other major concerns:

1. The definitions of "active" conformation of RadA in this manuscript (line 34) and "inactive" conformation in the paper Marie et al., 2017 is inappropriate. The binding of ATPγS or dTDP has nothing to do with the "active" or "inactive" states of a helicase.
2. The interaction between DNA and RadA or ComM is mediated by several domains, but only the interaction of DNA with one subunit was discussed in detail? Please also summarize the differences of the 6 subunits in interacting DNA.
3. Please provide a figure to better explain the working model summarized in the last section of Discussion.

4. Although categorized as translocases, RadA and ComM only captured dsDNA segments in the determined structures. Please comment on this inconsistency.

Referee #3:

Rosa et al report cryo-EM structures of two functionally related AAA+ ATPases involved in bacterial transformation and DNA branch migration during homologous recombination. The study unveils structurally analogous hexameric assemblies assembled via a Lon like hexamer and DNA translocating ATPase. Interestingly a large interdomain linker similarly aligns the Lon subunit in the quaternary structures relative to the ATPase cores, despite a rearranged secondary structure at the protein sequence level. The authors support the high-quality protein-DNA cryo-EM reconstructions with biochemistry probing the novel role for an arginine switch in regulating interdomain interactions, ATP hydrolysis and AAA+ function for these proteins. Overall, the data is very clearly presented and of high quality. The extended discussion is scholarly and appropriate. The paper is likely of interest to those studying AAA ATPases and DNA motor proteins in general. I have minor points for the authors consideration.

Points:

1. The discussion of the active site needs to be clarified:

Lin 224-225. Does E125 really activate the catalytic water in RadA? It does not appear appropriately positioned to do so. Do they mean E124? I think the diagram is mislabeled, or the text has these residues swapped. Do the authors observe solvent in their map proximal to the gamma phosphate? Is there a catalytic water present? What is the equivalent residue in ComM, E302? This is not discussed.

2. Line 252. Use gamma phosphate, rather than "terminal" phosphate?

3. Line. 252. His213 looks to be too far away to interact with the gamma phosphate? Is it positioned to align E124 to activate a water for hydrolysis instead?

4. A diagram (perhaps in figure 1) depicting the branch migration reactions to go with the introductory text would be useful for a general audience. There are no model figures anywhere in the paper, and this would help orient the reader and better place the structural biology in its biological context.

5. line 162. please provide a figure of density of distorted base pairing. Is there anything to explain why the DNA ends are disrupted.

6. Line168. There is no supplementary Figure 3 panel that makes sense with this statement?

7. line 183 184. Reference to figure 2b. Please label the density maps, and locate the Zn finger domain density that could not be modeled.

8. in Figure 2, please label the proteins by name to orient the reader.

9. Supplementary Table 3 is of poor quality.

10. Supplementary figure 7. The Zf density appears to stabilize the dodecamers. Can an alphfold model for the Zn be fit into that density? There is no mention of the role of the Zf in stabilizing the dodecamer. How does this compare to the MCM protein assemblies?

11.Line 221. This sentence is awkwardly worded.

12. Figure 3b and e. Please label 3' and 5' ends of DNA. It is hard to see the labels for RL1, H2I, Ps1B in these figures. The loops could be colored as in Supp Figure 5d to improve presentation. Where is A329 and A330 that are discussed (line 289)?

13. Line 268. Discussion of DNA contacts from G222 and R224. Is there a helix dipole-phosphate backbone interaction worth mentioning here?

14. Line 278. Is wording this passage "consistent with" more appropriate than "confirming" that ATP hydrolysis results in DNA release?

15. Line 295. The use of "lagging strand" terminology is confusing, at least to this referee. Maybe if you had a schematic of the DNA-protein complex with strands labeled to identify the strands this would be less confusing. Even so, lagging strand does not

make sense?

16. Figure 5a. It is difficult to see the interactions mediated by R301. A zoomed in interaction showing distances of R301 to E239 or E76 would be informative, and support the discussion better.

17. Line 613. Methods. Please report the molar concentration of the hexamers used for RadA and Comm. What is the molar ratio of protein to nucleic acid used for complex formation?

18. Line 619. Vitrobot not Vitroblot.

Referee #1:

The paper by Talachia Rosa and colleagues reports the cryoEM structures of RadA and ComM, two ATPase motor proteins responsible for branch migration of D-loop boundaries during natural transformation in bacteria, together with accompanying biochemical characterisation of structure-based mutants. The novelty of the present work consists principally in the detailed elucidation of the mechanism of DNA binding by RadA and ComM. Many questions remain concerning the mechanism of branch migration by these proteins but in my view the presented evidence represents a sufficiently significant advance to warrant publication in EMBOJ.

We thank referee 1 for her/his positive assessment of our manuscript and our work.

Overall, the data presented by the authors is impressive for extent and quality. I don't have major concerns, but I would like the authors to rewrite the points below in the Discussion. The English could also be improved (see a couple of examples below).

We have carefully checked and improved the clarity of the text throughout the manuscript.

Discussion

In general, the discussion of the mechanism of RadA loading onto dsDNA should be improved. The authors should highlight that it is highly unusual for a ring helicase/translocase to self load onto DNA, like they seem to imply. Both the MCM and DnaB helicases are loaded onto origin DNA by specific protein loaders. What are the mechanistic steps that would lead from DNA-free RadA to DNA-bound RadA ring, such as seen by the authors in their cryoEM work?

We have added a few sentences to clarify that we have not shown that RadA or ComM are able to self-load onto DNA *in vivo*: “RadA and ComM are both capable of self-loading on ssDNA and dsDNA *in vitro*. Whether they are capable of self-loading on their HR DNA substrates *in vivo* has not been firmly established. We have previously reported evidence for a functional interplay between pneumococcal RadA and RecA at three-way DNA junctions, where RecA assists RadA helicase activity (Marie *et al*, 2017). Potential interactions between ComM and other proteins *in vivo* are less documented”.

Related to this, the MCM helicase does not self-load onto origin DNA (line 420)! Surely the authors are aware that the Origin Recognition Complex is responsible for loading the MCM ring onto origin DNA.

We agree that our description of the loading mechanism of MCM on DNA was ambiguous in the discussion. We have amended the discussion to clarify this.

Again related to this, the authors should also mention that the use of relatively short linear DNA substrates might have caused RadA to be threaded onto the DNA, rather than self-loading in a manner that is reflective of its true mechanism.

We added a few sentences to discuss this specific point: “it is important to note that because we used short DNA fragments in this study, RadA could instead have been threaded along the DNA without the need to open the RadA ring during the process. However, we have previously shown that it is possible to obtain RadA hexamers loaded onto long circular

ssDNA molecules *in vitro* (Marie *et al*, 2017), a process that requires the opening of the ring.”

The observation that ComM can form double-hexamers on dsDNA is intriguing. Is it known whether ComM can branch migrate independently from RadA? And how do the authors envisage that RadA and ComM might cooperate in branch migration?

It was previously shown that ComM can branch migrate *in vitro* on its own, and thus independently from RadA and *vice versa*.

For ComM see in Nero TM, Dalia TN, Wang JC, Kysela DT, Bochman ML, Dalia AB. ComM is a hexameric helicase that promotes branch migration during natural transformation in diverse Gram-negative species. *Nucleic Acids Res.* 2018 Jul 6;46(12):6099-6111. doi: 10.1093/nar/gky343. PMID: 29722872; PMCID: PMC6158740.

For RadA see in Cooper DL, Lovett ST. Recombinational branch migration by the RadA/Sms paralog of RecA in *Escherichia coli*. *Elife.* 2016 Feb 4;5:e10807. doi: 10.7554/eLife.10807. PMID: 26845522; PMCID: PMC4786428.

In Gram-positive firmicutes species, there is no ComM homolog, which is specific to naturally transformable Gram-negative species. In Gram-negative species, RadA and ComM could both assist branch migration during homologous recombination. However, these two proteins are not expressed under the same conditions, with ComM being specifically expressed during competence and RadA being constitutively expressed. Nero *et al.* showed that RadA is not required for natural transformation in *V. cholerae*, suggesting that these two proteins are directed towards their specific role by additional factors in such bacterial species (a specific functional channeling directed by accessory proteins, or DNA substrate topology, etc...); i.e RadA acting in genome maintenance pathways and ComM in natural transformation.

Methods

1. Show SDS-PAGE of purified RadA and ComM.

These gels are now shown in supplementary figure 3

2. "CryoEM sample preparation" - give protein concentration in molarity, not mg/ml.
Done

Minor points

The authors occasionally refer in the text to the 'lagging strand' (for instance on line 270: "A second loop named RL2, which consists of residues 170 to 189, coordinates the lagging 3' to 5' strand."). It is not clear what they mean by lagging strand outside of DNA replication. Please clarify.

The nomenclature used, which was indeed based on the nomenclature used for DNA replication, was misleading. We now use a nomenclature based on the topology (5'-3' or 3'-5') of the DNA strand in interaction within the RadA or ComM hexamer from ATPase domain to Lon domain.

The English could be made more accurate to improve the readability of the manuscript. For

instance, consider the sentence on line 185: "Notably, although ComM and RadA have an inverted sequence order, they share identical topology between the ATPase and Lon domains". It's not the sequence, it's the arrangement of ATPase and Lon domains that is swapped. Also the structures show that they have identical reciprocal arrangements of ATPase and Lon domains, not identical topology. Please correct.

This sentence was indeed incorrect and has been corrected.

On line 166, "We overexpressed ComM from *L. pneumophila* in *E. coli*, attaching a C-terminal His6-tag." Please use 'fused to' rather than 'attaching'.

Corrected

Please write "AMPP-N-P" as AMPPNP.

Corrected

Referee #2:

RadA and ComM are important bacterial helicases involved in natural transformation with branch migration activity. In this manuscript, Rosa et al. report the structures of RadA and ComM in complex with their DNA substrates. These results would contribute to the understanding of the functions of RadA and ComM in branch migration. With this said, however, this study provides little insight into the mechanisms of these two helicases.

Moreover, the current version of the manuscript does not meet the criteria for being considered for acceptance, and should be largely revised. Most of the manuscript is on structural descriptions, which are too lengthy and hard to follow. Some descriptions appear inappropriate or lack support from structural information. The figures were inadequately prepared and did not effectively support the structural descriptions. Please refer to previously published structural papers in the EMBO J to improve the manuscript and figures.

We thank referee #2 for her/his detailed evaluation of our manuscript that greatly helped us to improve our manuscript.

We fully agree with referee #2 that our figures did not meet publishable standards and that they should be drastically improved.

As a result, all the main figures and extended views have been thoroughly revised. We have also modified the main text to improve readability and clarity. We have also modified the discussion to emphasize the novelty of our results and what they bring to the field.

Here are some examples for improvement:

(1) Line 146-147, the flexibility of subunit F was not shown in Figure S1.

Indeed. Figure S1 shows that the resolution of the map corresponding to this region is lower (maybe due to flexibility of this region). We have corrected the text.

(2) Line 151-153, where is the N-terminal arm in Figure 2a? All the domains, motifs or residues mentioned in the text should be clearly labeled in the supporting figures.

This is now shown in the new version of the figure.

(3) Line 153-154, where is the ZF domain in Figure S1.

The ZF domain is completely invisible on the map. Therefore, we cannot show where it should be in the map. We remove the reference to Figure S1 from the text.

(4) Line 161-163, what does "DNA-B" mean? The B form DNA?

We referred to the B-form of DNA (B-DNA). This has been corrected in the text.

The descriptions of some concepts or definitions should be accurate. Please show the structural evidence for the "distorted or disrupted" downstream DNA region. In addition, when the authors assumed that the downstream DNA around the Lon domain is distorted or disrupted, why did they propose that RadA captured the dsDNA end (line160-161)?

We agree that we cannot claim that the DNA in the Lon region of ComM is disrupted or distorted. The resolution of the map corresponding to this part of the DNA drops drastically, which is more a sign of the flexibility of this region. We have corrected this sentence in the text.

(5) Line 168-169, where is the 50 kDa monomer protein in Figure S3.

Chromatograms and SDS-PAGE corresponding to these experiments have been included in supplementary figure 3.

(6) Line 180-181, the sentence discusses the domain similarity between the AAA+ ATPase domain of ComM and MCM. It raises the question as to why the authors did not present a superimposition of the ComM domain with that of MCM, similar to what was done for the domain comparison of ComM, RavA, and Chelatase.

We added the MCM in the overlay shown in the extended view Figure 2e.

(7) Line 184, where is the ZF domain?

The densities corresponding to the SF domain are now shown in Figure 2.

(8) Line 187, label all the mentioned residues in figures.

Done in the new version of the Figure 1a.

(9) Line 191, label the chelatase domains.

Done in the new version of Figure 1b and 1c.

(10) In lines 256-259, the description lacks sufficient evidence. Please provide a detailed explanation of the molecular mechanism by which histidine stacks with the short helix and how the formation of the helix pushes forward the RL1 loop. Otherwise, consider removing this sentence.

We now use the conditional in this sentence to emphasise that it is a hypothesis rather than a result.

"In RadA, we propose that the stacking of these two histidine could structure a short helix at the N-terminus of the KNRF motif (R224 to H228), which in turn would push the adjacent loop (RadA loop 1 or RL1, residues 213 to 223) forward to promote DNA binding (Figure 3c)."

(11) Line 295-265, as the interaction between Lon domain and DNA was not observed in the structure, is there direct evidence to support this description?

No. There is no direct evidence. But we propose that this domain could also be involved in DNA binding as shown for homologous RadA protein.

(12) Line 269-273, where are the RL2 and residues 170-189?

It is now shown in the new version of the Figure 3.

(13) Line 277-278, this description lacks a supporting figure.

This is now shown in the Figure 3 (panel e).

(14) Line 287-289, please show these interactions in detail.

We have toned down our description of the molecular interactions of ComM with DNA. In fact, the resolution is not sufficient to place the side chains of the residues in these regions accurately. We only suggest possible interactions between ComM and DNA.

(15) Line 290-292, please present the "half size" in figures.

See above.

(16) Line 293-294, figure 3f didn't support this conclusion.

See above.

(17) Line 298, what does the lagging strand mean?

This point was raised by Referee #1. We changed the nomenclature for DNA strands to avoid ambiguity.

(18) Line 327, the characterization of the R301-mediated domain interaction as a molecular switch is not justified. While the interaction of R301 in the Lon domain with E239/E76 is crucial for inter-domain interaction, it likely represents only a fraction of the intricate interactions between the Lon and ATPase domains. It is an exaggeration to claim R301 interaction as a molecular switch. The other residues contributing to the interface should also be described.

We agree. We have removed the mention of a molecular switch in the text.

However, for clarity, we have decided not to show other residues involved in the contacts between the ATPase and Lon domains.

(19) Line 339-340, what does this sentence mean?

We have removed this sentence.

(20) Line 226: G100 could not be found in figure 3a.

We do not mention G100 in the text anymore.

(21) Line 234: the PDB 6MII cannot be found in S5.

We corrected the text.

(22) In figure 2c, ATPyS should be ATPγS. The display mode is not very clear in figure2.

We have drastically modified Figure 2 and hope that it is better now.

Other major concerns:

1. The definitions of "active" conformation of RadA in this manuscript (line 34) and "inactive" conformation in the paper Marie et al., 2017 is inappropriate. The binding of ATPγS or dTDP has nothing to do with the "active" or "inactive" states of a helicase.

We agree. We have removed these definitions from the text.

2. The interaction between DNA and RadA or ComM is mediated by several domains, but only the interaction of DNA with one subunit was discussed in detail? Please also summarize the differences of the 6 subunits in interacting DNA.

We added panels in Figure 3 (g and h) and corresponding text in the manuscript to summarize these differences.

3. Please provide a figure to better explain the working model summarized in the last section of Discussion.

We added the new figure 6.

4. Although categorized as translocases, RadA and ComM only captured dsDNA segments in the determined structures. Please comment on this inconsistency.

We and others initially proposed that RadA and ComM could be helicases translocating along ssDNA based on their homology with DnaB and MCM replicative helicases, respectively, and based on previous and pioneering in vitro analysis of their DNA interaction and remodelling activities. .

Capturing these two proteins on dsDNA opens the possibility that they could be dsDNA translocases. This point is discussed in the last section of the discussion.

Referee #3:

Rosa et al report cryo-EM structures of two functionally related AAA+ ATPases involved in bacterial transformation and DNA branch migration during homologous recombination. The study unveils structurally analogous hexameric assemblies assembled via a Lon like hexamer and DNA translocating ATPase. Interestingly a large interdomain linker similarly aligns the Lon subunit in the quaternary structures relative to the ATPase cores, despite a rearranged secondary structure at the protein sequence level. The authors support the high-quality protein-DNA cryo-EM reconstructions with biochemistry probing the novel role for an arginine switch in regulating interdomain interactions, ATP hydrolysis and AAA+ function for these proteins. Overall, the data is very clearly presented and of high quality. The extended discussion is scholarly and appropriate. The paper is likely of interest to those studying AAA ATPases and DNA motor proteins in general. I have minor points for the authors consideration.

We thank referee #3 for her/his favorable evaluation of our manuscript and detailed feedback.

Points:

1. The discussion of the active site needs to be clarified: Lin 224-225. Does E125 really activate the catalytic water in RadA? It does not appear appropriately positioned to do so. Do they mean E124? I think the diagram is mislabeled, or the text has these residues swapped. Do the authors observe solvent in their map proximal to the gamma phosphate? Is there a catalytic water present? What is the equivalent residue in ComM, E302? This is not discussed.

There was indeed an error in our text. The glutamate positioned to activate the catalytic water is E124. We have corrected this. We also mention that the catalytic water is not visible in our map (not enough resolution).

The corresponding residue in ComM is indeed E302. We now mention that it would be appropriate to activate the catalytic water.

2. Line 252. Use γ -phosphate, rather than "terminal" phosphate?

This has been corrected.

3. Line. 252. His213 looks to be too far away to interact with the gamma phosphate? Is it positioned to align E124 to activate a water for hydrolysis instead?

This is another error that has been corrected in the revised version of the manuscript.

4. A diagram (perhaps in figure 1) depicting the branch migration reactions to go with the introductory text would be useful for a general audience. There are no model figures anywhere in the paper, and this would help orient the reader and better place the structural biology in its biological context.

Such diagram, absent in the original version, is now provided in Figure 6. We don't feel it is necessary to provide another figure.

5. line 162. please provide a figure of density of distorted base pairing. Is there anything to explain why the DNA ends are disrupted.

This is discussed above (see referee #2).

6. Line168. There is no supplementary Figure 3 panel that makes sense with this statement?

Chromatograms and SDS-PAGEs have now been added.

7. line 183 184. Reference to figure 2b. Please label the density maps, and locate the Zn finger domain density that could not be modeled.

This is now clearly visible in Figure 2.

8. in Figure 2, please label the proteins by name to orient the reader.

Done.

9. Supplementary Table 3 is of poor quality.

We now provided a table of greater quality.

10. Supplementary figure 7. The Zf density appears to stabilize the dodecamers. Can an alphafold model for the Zn be fit into that density? There is no mention of the role of the Zf in stabilizing the dodecamer. How does this compare to the MCM protein assemblies?

The ZF domain for ComM is predicted with very low confidence by AF2 or AF3. So we could not dock any reliable structure prediction of this part of the protein in the map. The size of the density fits well with a domain of this size.

As described in the discussion, the dodecamerisation of MCM and ComM happens in opposite interfaces (ATPase domain for ComM and N-terminal domain in MCM).

11. Line 221. This sentence is awkwardly worded.
Corrected.

12. Figure 3b and e. Please label 3' and 5' ends of DNA. It is hard to see the labels for RL1, H2I, Ps1B in these figures. The loops could be colored as in Supp Figure 5d to improve presentation. Where is A329 and A330 that are discussed (line 289)?

We now provide a new version of the figure in which the 5' and 3' ends of the DNA are labelled in some panels.

13. Line 268. Discussion of DNA contacts from G222 and R224. Is there a helix dipole-phosphate backbone interaction worth mentioning here?

Sorry, we do not understand what interactions the referee #3 is referring to.

14. Line 278. Is wording this passage "consistent with" more appropriate than "confirming" that ATP hydrolysis results in DNA release?

This has been corrected accordingly.

15. Line 295. The use of "lagging strand" terminology is confusing, at least to this referee. Maybe if you had a schematic of the DNA-protein complex with strands labeled to identify the strands this would be less confusing. Even so, lagging strand does not make sense?

Referee #1 also asked us to correct this. We now changed to a nomenclature based on the polarity of the DNA relative to the hexamer.

16. Figure 5a. It is difficult to see the interactions mediated by R301. A zoomed in interaction showing distances of R301 to E239 or E76 would be informative, and support the discussion better.

We modified the figure 5 accordingly.

17. Line 613. Methods. Please report the molar concentration of the hexamers used for RadA and Comm. What is the molar ratio of protein to nucleic acid used for complex formation?

Done

18. Line 619. Vitrobot not Vitroblot.

This is now corrected.

Dear Remi,

Thank you for submitting a revised version of your manuscript. I sincerely apologise for the protracted assessment process due to delays in referee report submission. We have now received input from two of the original reviewers, who find that their previous concerns have been addressed satisfactorily. Therefore, there now remain only a few editorial points that need addressing before I can extend official acceptance of the manuscript:

1. Please submit up to five keywords.
2. Please check that the funding information is correct and identical both in the manuscript and our online system.
3. Please submit a complete author checklist, which you can download from our author guidelines (<https://www.embopress.org/pb-assets/embo-site/EMBO%20Press%20Author%20Checklist-1642513524327.xlsx>). Please insert information in the checklist that is also reflected in the manuscript. The completed author checklist will also be part of the Review Process File.
4. We are missing the ORCID iD for the co-corresponding author Patrice Polard. In order to link the ORCID iD to the account in our manuscript tracking system, the author in question has to do the following:
 - Click the 'Modify Profile' link at the bottom of your homepage in our system.
 - On the next page you will see a box halfway down the page titled ORCID*. Below this box is red text reading 'To Register/Link to ORCID, click here'. Please follow that link: you will be taken to ORCID where you can log in to your account (or create an account if you don't have one)
 - You will then be asked to authorise Wiley to access your ORCID information. Once you have approved the linking, you will be brought back to our manuscript system.Unfortunately, we cannot do this linking on the author's behalf for security reasons.
5. Please add a "Disclosure and competing interests statement" section. Further info: <https://www.embopress.org/page/journal/14602075/authorguide#conflictsofinterest>.
6. Figures need to be cited in sequential order: e.g. Fig 2A is called out before Fig 1B, Fig 4E is called out before Fig 3C; there is a callout for a Fig 1D (after Fig 3), which does not exist. Video 4 is called out before Video 3 and callout for Video 5 is missing. Callouts for EV figures are missing.
7. The legends of the EV figures should be under the heading "Expanded View Figure Legends", and callouts/legend titles should be updated to "Figure EV1" and "Figure EV2".
8. Supplementary videos are mentioned, but they are not uploaded. Please rename the movies into Movie EV1-EV6 and update the callouts accordingly. The legends should be removed from the manuscript text file and zipped with each movie file. Further information is available here: <https://www.embopress.org/page/journal/14602075/authorguide#expandedview>
9. In the Appendix, please add page numbers and a brief table of contents. Please update the nomenclature to Appendix Figure S1, etc. and add the Appendix Figure and Table legends to the Appendix file.
10. In the Data Availability section, please add a resolvable links to the deposited datasets. More information about the format of this section can be found here: <https://www.embopress.org/page/journal/14602075/authorguide#dataavailability>.
11. Our data editors have flagged the following issues in figure legends that need correcting:
 - Please note that the figure panel EV 1f is not labeled in the manuscript - please correct.
 - Please define the error bars in the legends of figures 5c, e.
12. Please upload the source data files for Figure 5 in one folder, labelled according to the corresponding figure panels to make sure they are correctly assigned during the typesetting process.
13. Papers published in The EMBO Journal are accompanied online by a 'Synopsis' to enhance discoverability of the manuscript. It consists of A) a short (1-2 sentences) summary of the findings and their significance, B) 3-4 bullet points highlighting key results and C) a synopsis image that is 550x300-600 pixels large (width x height, jpeg or png format). You can either show a model or key data in the synopsis image. Please note that the image size is rather small and that text needs to be readable at the final size. Please send us this information together with the revised manuscript.

With best wishes,

leva

leva Gailite, PhD
Senior Scientific Editor
The EMBO Journal
Meyerhofstrasse 1

D-69117 Heidelberg
Tel: +4962218891309
i.gailite@embojournal.org

We realize that it is difficult to revise to a specific deadline. In the interest of protecting the conceptual advance provided by the work, we recommend a revision within 3 months (17th Nov 2024). Please discuss the revision progress ahead of this time with the editor if you require more time to complete the revisions. Use the link below to submit your revision:

Referee #1:

I have read the revised manuscript by Fronzes and colleagues. I find that the manuscript is now sufficiently improved to warrant publication.

The discussion of the RadA and ComM translocation mechanisms repeats points already made in the Results and could be fruitfully made shorter.

Please review the text and figures carefully, there are several typos to correct. Also please consider having the manuscript reviewed by a native-speaker colleague to improve the English, which is occasionally clumsy.

Referee #3:

The authors have extensively addressed the referee critiques.

All editorial and formatting issues were resolved by the authors.

Dear Rémi,

Thank you for addressing most of the final editorial issues. I sincerely apologise for the delay in the processing of your manuscript due to my absence from the office at the first half of September and the resulting backlog. I am now pleased to inform you that your manuscript has been accepted for publication.

Before we forward your manuscript to the publishers, I would still need the information on the nature of the scale bars in figure panels 5C and 5E - e.g., if they represent standard deviation of SEM. I can add this information in the figure legend for you.

I will also look into the synopsis text that you kindly provided and will let you know if any edits to the journal style are needed.

If you have any questions, please do not hesitate to contact the Editorial Office. Thank you for this contribution to The EMBO Journal and congratulations on a nice study!

Best wishes,

leva

leva Gailite, PhD
Senior Scientific Editor
The EMBO Journal
Meyerhofstrasse 1
D-69117 Heidelberg
Tel: +4962218891309
i.gailite@embojournal.org
